# Mathematical modeling unveils the timeline of CAR-T cell therapy and macrophage-mediated cytokine release syndrome

**Daniela S. Santurio[1], Luciana R. C. Barros[2], Ingmar Glauche[3], Artur c. Fassoni** [3,4]*

**1** Certara UK Ltd., Sheffield, United Kingdom, **2** Instituto do Câncer do Estado de São Paulo, São Paulo, Brazil, **3** Institute for Medical Informatics and Biometry, Technische Universität Dresden, Dresden, Germany, **4** Instituto de Matemática e Computação, Universidade Federal de Itajubá, Itajubá, Brazil

* fassoni@unifei.edu.br

**Data availability statement:** The patient data that support the findings of this study were

## Abstract

Chimeric antigen receptor (CAR)-T cell therapy holds significant potential for cancer treatment, although disease relapse and cytokine release syndrome (CRS) remain as frequent clinical challenges. To better understand the mechanisms underlying the temporal dynamics of CAR-T cell therapy response and CRS, we developed a novel multi-layer mathematical model incorporating antigen-mediated CAR-T cell expansion, antigen-negative resistance, and macrophage-associated cytokine release. Three key mechanisms of macrophage activation are considered: release of damage-associated molecular patterns, antigen-binding mediated activation, and CD40-CD40L contact. The model accurately describes 25 patient time courses with different responses and IL-6 cytokine kinetics. We successfully link the dynamic shape of the response to interpretable model parameters and investigate the influence of CAR-T cell dose and initial tumor burden on the occurrence of cytokine release and treatment outcome. By disentangling the timeline of macrophage activation, the model identified distinct contributions of each activation mechanism, suggesting the CD40-CD40L axis as a major driver of cytokine release and a clinically feasible target to control the activation process and modulate cytokine peak height. Our multi-layer model provides a comprehensive framework for understanding the complex interactions between CAR-T cells, tumor cells, and macrophages during therapy.

## Author summary

CAR-T cell therapy uses specially engineered immune cells to target and destroy cancer cells. While this treatment has shown great promise, many patients experience serious side effects, particularly cytokine release syndrome (CRS), a condition that can cause

collected from existing literature [1,32,56–58]. These data and all author-generated code supporting the results of this study are available on Zenodo at link https://doi.org/10.5281/zenodo.14501736.

**Funding:** ACF was supported by Alexander von Humboldt Foundation and Coordenação de Aperfeiçoamento de Pessoal de Nível Superior - Brasil (CAPES). The funders had no role in study design, data collection and analysis, decision to publish, or preparation of the manuscript.

**Competing interests:** The authors have declared that no competing interests exist.

severe inflammation and organ damage. To better understand these challenges, we developed a comprehensive mathematical model that simulates the interactions between CAR-T cells, tumor cells, and macrophages—immune cells that play a role in driving excessive cytokine release. Our model successfully reproduced real patient data across different tumor types and therapy outcomes. This allowed us to simulate how factors such as CAR-T cell dose and initial tumor size influence the therapy's effectiveness and the likelihood of side effects. Notably, our model identified a specific interaction between CAR-T cells and macrophages as a key driver of cytokine release. This finding suggests that targeting this pathway could reduce the risk of severe side effects, potentially making CAR-T cell therapy safer and more effective. Our work provides a framework that can be used to optimize CAR-T cell therapy and improve patient outcomes.

## Introduction

Immunotherapy with chimeric antigen receptor T (CAR-T) cells is a groundbreaking approach that harnesses the power of the immune system to fight hematological and non-hematological malignancies [1]. Genetic engineering enables T cells to express chimeric antigen receptors (CARs), enhancing their ability to specifically recognize and eliminate cancer cells. This remarkable efficacy in treating certain types of blood cancers made CAR-T cell therapy a promising avenue in oncology, with new CAR designs entering clinical trials for several indications, including solid tumors [2].

The short-term response to CAR-T cell therapy is characterized by typical multiphasic dynamics, with the initial distribution phase being marked by a decline of injected cells due to their death and migration to tumor site [3–5]. The subsequent expansion phase encompasses tumor-killing and a massive proliferation of CAR-T cells mediated by antigen recognition. In the following contraction phase, exhausted CAR-T cells are eliminated, and a fraction of long-lived memory CAR-T cells remain in the final persistence phase. Using mathematical models we and others showed how this multiphasic dynamics emerge from the interaction of different CAR-T cell phenotypes with tumor cells [3–5].

While CAR-T cell therapy is as a paradigm-shifting modality for cancer treatment, it faces substantial challenges, with 30-60% of patients experiencing recurrence within one year post-treatment [6]. Its effectiveness is linked to biological mechanisms of CAR-T cells, such as expansion, cytotoxicity, and memory formation, all of which are influenced by antigen recognition. Consequently, the density of antigens presented by tumor cells constitutes a pivotal factor for therapy resistance and transient antigen loss under treatment pressure or permanent antigen loss due to mutations account for typical resistance mechanisms [7].

Besides resistance, major obstacles for CAR-T cell therapy are side effects, including cytokine release syndrome (CRS) and immune effector cell-associated neurotoxicity syndrome (ICANS) [8]. Affecting 13%–26% of patients, severe CRS is the most prevalent adverse event, with symptoms typically emerging 1–7 days post-infusion, commonly before the CAR-T cell peak [9,10]. CRS is characterized by fever, hypotension, and respiratory insufficiency and is associated with rising levels of cytokines, including IL-6, IL-10, IL-8, interferon-$\gamma$ (IFN-$\gamma$) and granulocyte–macrophage colony-stimulating factor (GM-CSF) [11,12]. In many cases CRS severity is tied to tumor burden and sometimes to CAR-T cell dose [13]. CRS also influences patient survival, and severe CRS (grade 3-5) correlated with lower survival in multiple myeloma (MM) and acute lymphoid leukemia (ALL) patients [11,12].

Preclinical studies suggest that CRS results from complex interactions between CAR-T and host cells, with macrophages as a primary source of released cytokines [8,14–16]. Three

mechanisms have been identified as major drivers of macrophage activation underlying CRS [8]. First, upon antigen-binding-mediated activation, CAR-T cells release inflammatory signals such as IFN-$\gamma$ and GM-CSF that induce macrophage activation in a contact-independent manner. Second, following tumor cell targeting by CAR-T cells, damage-associated molecular patterns (DAMPs) released by pyroptotic tumor cells enhance macrophage activation through pattern-recognition receptors. Third, the interaction of CD40 expressed on activated macrophages with CD40L expressed on CAR-T cells further promotes macrophage activation in a contact-dependent manner.

Different aspects of CAR-T cell therapy were investigated by modeling approaches [17–20] with a few focusing on CRS [21–24]. However, the critical role of macrophages on cytokine release was rarely addressed [25]. It is currently unclear how the different macrophage-associated mechanisms influences the extent and the temporal dynamics of cytokine release, although these aspects are essential for clinical implementation of countermeasures. With refined models, therapeutic interventions can be tested, as for example modulating CAR-T cell dose or changes in preconditioning to reduce the initial tumor burden. The models can also describe supportive administration of different corticosteroids or antibodies to inhibit macrophage activation and cytokine release.

Here, we develop a multi-layer model of CAR-T cell therapy encompassing antigen-negative resistance and macrophage-associated cytokine release. The model consistently describes time courses of 25 patients with different malignancies and tumor responses. Explicitly including three mechanisms of macrophage activation (DAMPs release, antigen-binding, and CD40 contact), the model also fits to cytokine time courses of 15 patients. We dissect the different phases of macrophage activation, showing that each mechanism occurs at a different time point during therapy response, with different contributions. Our results provide insights on macrophage-mediated cytokine release and elicit testable hypotheses on therapeutic interventions to mitigate toxicity in CAR-T cell therapy.

## Results

### The model accurately reproduces diverse patient-specific responses

The fundamental layer of our modeling approach is a minimal model describing CAR-T cell multiphasic dynamics and antigen-positive tumor responses (Fig 1A, Eqs (1)–(5)). This layer encompasses three CAR-T cell populations, namely injected, expander, and persister cells, and a population of antigen-positive tumor cells. The underlying model emerges as simplification from our previous model [5] with a reduction from 19 to 12 parameters, now integrating concepts of T-cell turnover [26] and plasticity [27,28] which explain transitions between CAR-T cell phenotypes as functional responses to antigen-binding.

Our modeling approach is based on the idea that the T-cell response is driven by activated, expanding cells that eliminate tumor cells, and persistent, long-lived memory cells that form a silent reservoir, which can be reactivated to maintain sustained remission [29]. Instead of fixed progenitor relationships between effector and memory T cells, our model assumes phenotypic plasticity between these cell types depending on the available antigen. Given that central memory and effector memory cells can also proliferate [30,31], we group these phenotypes into a single population termed CAR-T expander cells, while long-term memory cells with low cytotoxicity are named CAR-T persister cells, consistent with the modeling framework proposed in [29]. The switch between these phenotypes depends on the antigen load. High antigen levels activate injected CAR-T cells, induce the proliferation of CAR-T expanders, and lead to conversion of persister cells into expander cells and inhibit the formation of persister cells.

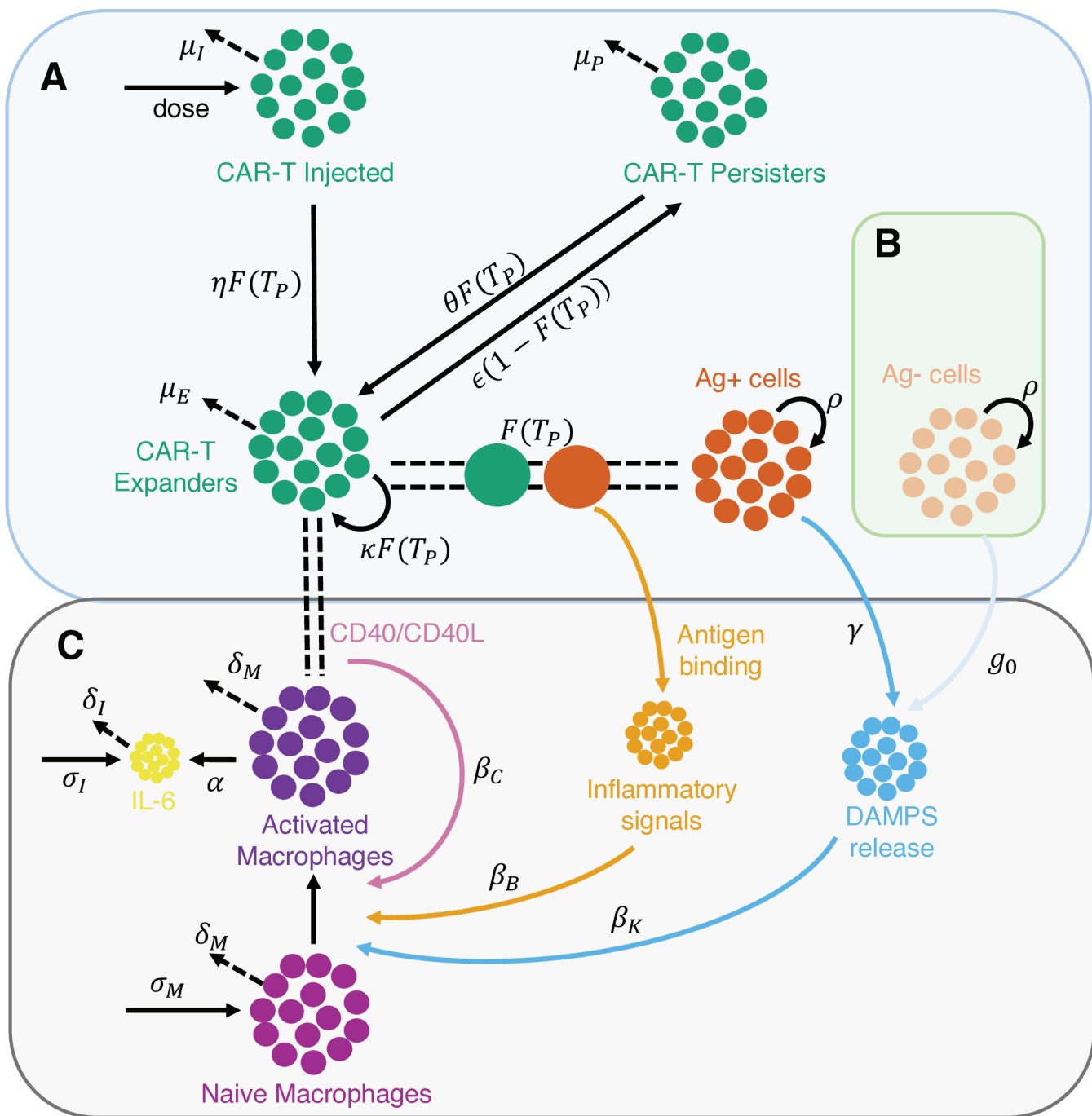

**Fig 1. Model schematic description**. **A** Model for CAR-T cell therapy considering injected ($C_I$), expander ($C_E$), and persister ($C_P$) CAR-T cells, and antigen-positive (Ag+) tumor cells ($T_P$). The phenotypic differentiation of CAR-T cells is driven by antigen recognition on the surface of tumor cells ($F(T_P)$). **B** The model is extended to describe patients who presented antigen-negative tumor relapses by including a compartment of antigen-negative (Ag-) tumor cells ($T_N$). **C** The same model can be extended to describe macrophage-mediated cytokine release considering three different activation mechanisms: antigen-binding (upon antigen-recognition, activated CAR-T cells release inflammatory cytokines and molecules that activate naive macrophages, orange arrow); DAMPs-release (CAR-T-mediated killing of the tumor leads to the release of damage-associated molecular patterns (DAMPs) that promote macrophage activation, blue arrow); CD40-contact (in a contact-dependent manner, activated macrophages expressing CD40 bind to CD40 ligand expressed by CAR-T cells and promote further macrophage activation (pink arrow).

Fitting the model (see Methods) to n = 19 time courses of CAR-T-treated patients with different malignancies (Figs 2, S1 and S2), shows that the model accurately describes both the CAR-T cell multiphasic dynamics as well as the different types of tumor responses.

Tumor relapse occurs in 30-60% of patients within one year after CAR-T therapy [6] and many relapsed patients present antigen-negative tumor cells, which are more resistant to CAR-T cytotoxicity. To account for those cases, we add a second layer to the basic model, consisting of a population of antigen-negative tumor cells (Fig 1B, Eq (7)). Due to the lack of antigen, we assume that these cells do not contribute to the antigen-mediated phenotypic changes in CAR-T cells. However, they are subject to a minimal cytotoxic effect by CAR-T cells due to the bystander effect and CAR-T cell endogenous T-cell receptor [7,33]. The model

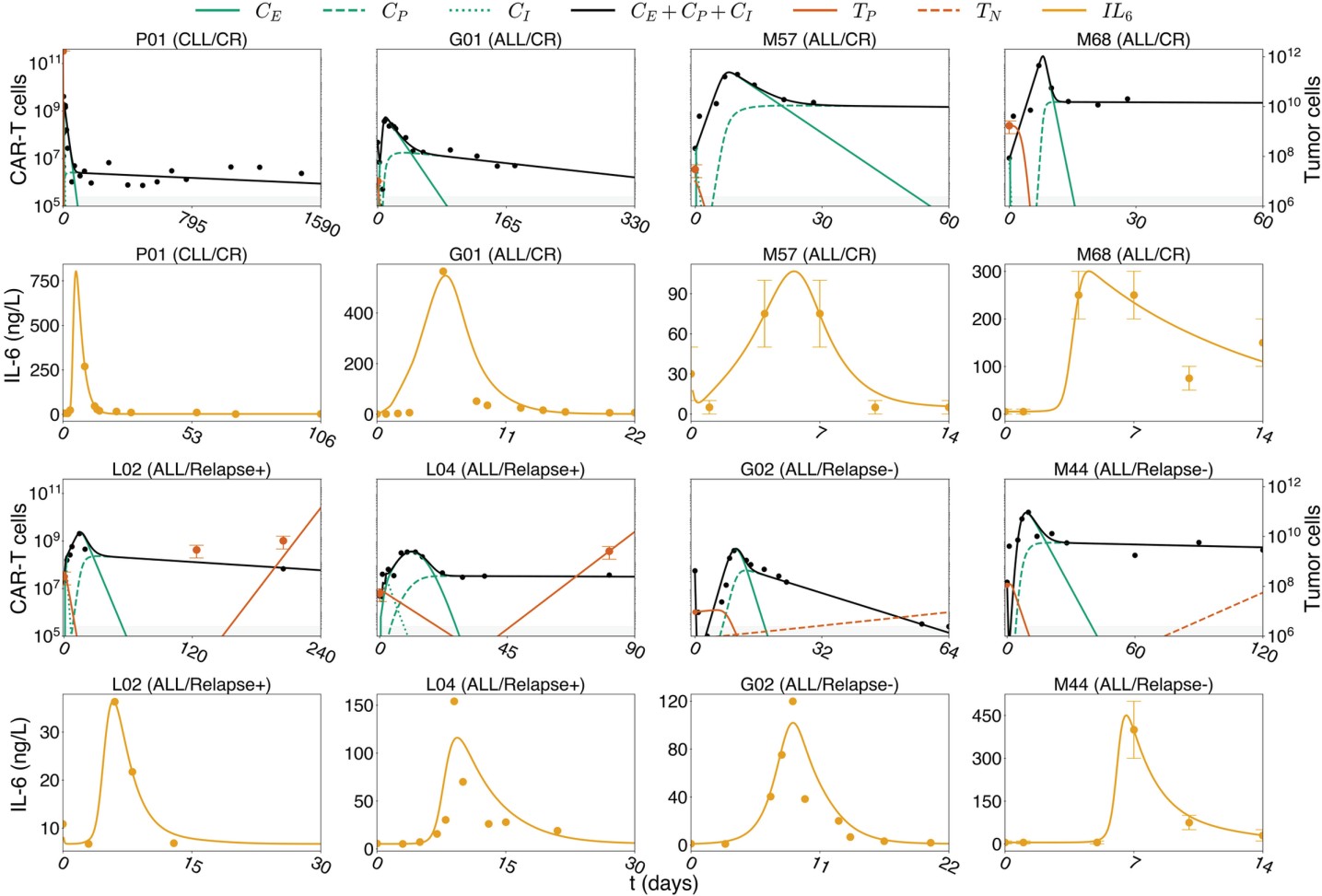

**Fig 2. Model simulations.** Model fits for selected patients that showed response to therapy (P01, G01, M57, and M68) or relapse either of antigen-positive tumor cells (L02, L04) or antigen-negative tumor cells (G02, and M44). See S1–S4 Figs for all 25 patients and Methods for details on parameter estimation. Although patients M57 and M68 were simulated until their PFS day, plots show their dynamics for 60 days for clarity. Tumor cell error bars represent the range of WBCs used in scaling while error bars for IL-6 in patients M44, M57, and M68 encompass the range reported in [32]. The CAR-T cell detection threshold of $2.5 \times 10^5$ cells is represented by the gray shaded area.

was fitted to n=6 time courses of antigen-negative relapsed patients (Figs 2 and S3). We confirm that the model quantitatively and qualitatively reflects the dynamics of antigen-negative relapses.

To detect early indications of side effects in CAR-T cell therapy, the concentrations of critical cytokines are usually monitored in parallel to treatment response, specifically IL-6. Typically, these cytokines follow a predictable kinetics: starting from a baseline concentration, they often peak before the CAR-T cells, and then return to baseline levels. It is increasingly evident that macrophages play a pivotal role in the release of cytokines in response to CAR-T cell therapy [8,14,16]. To quantitatively describe macrophage-mediated cytokine kinetics and gain a deeper understanding on the mechanisms underlying CRS, we add a third layer to our model including three compartments, namely naive macrophages, activated macrophages and IL-6 blood concentration (Fig 1C, Eqs (8–11)). We focus our attention on IL-6 as more time course data is available.

The main assumption of the model is that macrophage activation is driven by three different, well-described mechanisms: (i) the release of DAMPs by the killing of tumor cells, (ii) the release of inflammatory signals by CAR-T cells upon antigen recognition, and (iii) the contact of activated macrophages and CAR-T cells through the CD40-CD40L axis. In our model we further assume basal influx/production and death/decay rates for monocytes and cytokines, and that IL-6 is released by activated macrophages. We collected additional IL-6 time courses for a subset of 15 patients for which also CAR-T cell time courses were available. The model is fitted with excellent agreement between data and simulations (Figs 2 and S4).

We conclude that our modular model structure accurately describes the main dynamical aspects of CAR-T cell therapy, explicitly accounting for CAR-T cell multiphasic dynamics, antigen-positive and negative relapses, and the occurrence of macrophage-mediated CRS.

## Different functional mechanisms shape the dynamics of therapy response

The response to CAR-T cell therapy unfolds over time, characterized by distinct phases in CAR-T cell dynamics. Mathematical approaches are suited to identify the functional mechanisms driving each response phase [5]. The fact that both CAR-T and tumor cell numbers change rapidly by different orders of magnitude after CAR-T cell infusion translates into mathematical arguments that allow simplifying the model equations for each phase and then identifying leading order effects at different timescales. As detailed in the Methods section, each phase can be approximated by an exponential function that is represented on the logarithmic scale as a straight line with characteristic slopes related to different model parameters and reflecting patient-specific values.

The validity of these approximations is confirmed by comparing the maximum and minimum values achieved by the slopes of the model-fitted solutions during each phase with the values of each parameter assigned to each exponential phase (Fig 3A–3E). Together, these results show that the four phases of CAR-T cell dynamics are driven by distinct mechanisms and cellular phenotypes. The distribution phase is dominated by the decay of injected CAR-T cells, with the minimum slope of model solutions closely correlated to the decay rate $-\mu_I$. The subsequent expansion phase is driven by CAR-T expanders, with the maximum slope correlated to the net effect between their expansion and reduction due to death and exhaustion, represented as $\kappa - \mu_E$. The contraction phase is characterized by the exhaustion and death of CAR-T expander cells, with the minimum slope correlated to their depletion rate $-\mu_E$. The final persistence phase is dominated by the presence of residual persistent CAR-T cells, with maximum slope closely correlated to their death rate $-\mu_P$.

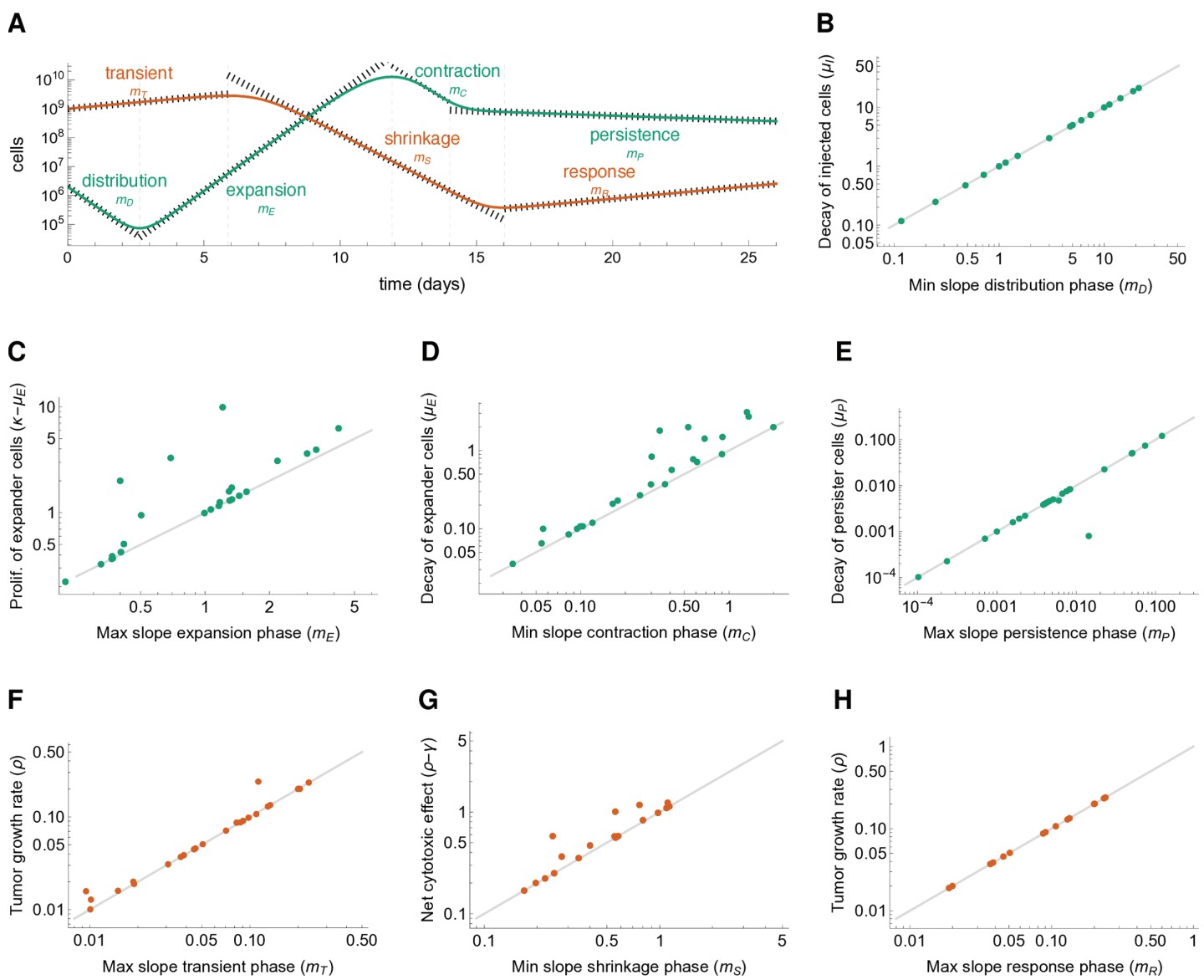

**Fig 3. Identifying mechanisms driving therapy phases.** **A** For each patient model fit, the distinct phases for CAR-T (green) and tumor (red) cell responses are identified and the characteristic slope is calculated and compared with the patient-specific mechanistic parameter driving the corresponding phase. **B-H** Comparison between the calculated slope and the mechanistic parameter for each phase and each patient. **B** Distribution phase: the calculated slope is the minimum (negative) slope of CAR-T cells, the mechanistic parameter is $-\mu_I$. **C** Expansion phase: the calculated slope is the maximum (positive) slope of CAR-T cells, the mechanistic parameter is $\kappa - \mu_E$. **D** Contraction phase: the calculated slope is the minimum (negative) slope of CAR-T cells, the mechanistic parameter is $-\mu_E$. **E** Persistence phase: the calculated slope is the maximum (negative) slope of CAR-T cells, the mechanistic parameter is $-\mu_P$. **F** Transient phase: the calculated slope is the maximum (positive) slope of tumor cells, the mechanistic parameter is $\rho$. **G** Shrinkage phase: the calculated slope is the minimum (negative) slope of tumor cells, the mechanistic parameter is $\rho - \gamma$. **H** Response phase: the calculated slope is the maximum (positive) slope of tumor cells for relapse patients only, the mechanistic parameter is $\rho$.

We also identify three phases for tumor responses (Fig 3F—3H). The first is a short and sometimes missing transient phase characterized by tumor growth while the injected CAR-T cells are being activated. Its maximum slope is correlated to the tumor growth rate $\rho$. The second phase is the shrinkage phase, characterized by a strong decay in the tumor burden due to the CAR-T-mediated killing. Its minimum slope is correlated to the difference between the tumor growth rate and CAR-T cytotoxic effect, $\rho - \gamma$. The last phase is the response phase,

which may be either a re-expansion leading to relapse with maximum slope closely correlated to the tumor growth rate $\rho$, or tumor extinction. The final outcome depends on an intricate balance of all pro- and anti-tumor parameters and whether or not the CAR-T cell-mediated killing reduced the tumor burden below a residual disease level.

The identification of model parameters driving the various phases of therapy response in clinical data not only enhances our mechanistic understanding but also supports a biologically motivated, stepwise parameter estimation approach. In this procedure (see Methods for details), the slope values obtained from patient time courses (denoted as phenomenological parameters) are used to inform the values of the corresponding mechanistic parameters, ensuring that the best-fit values are biologically grounded and interpretable. As we demonstrate below, the model fitting is further complemented by identifying other parameters governing other shape features of the model solutions.

In summary, we could successfully and uniquely link the different phases of the CAR-T and tumor cell response to the underlying mechanistic parameters of our mathematical model.

## Analysis of CAR-T and tumor cell dynamics enables informed parameter estimation

The preceding analysis enhances our understanding of the multiphasic dynamics following CAR-T cell administration and reveals a mapping between six characteristic slopes ($m_D, m_E, m_C, m_P, m_S, m_P$, Fig 3) and six model parameters combinations ($\mu_I, \kappa - \mu_E, \mu_E, \mu_P, \gamma, \rho$), where each slope is approximated by the respective combination at first-order. In terms of parameter estimation, this allows for an informed initial guess for these parameters. Here, $\mu_I, \mu_E$ and $\mu_P$ are the decay rates of injected, expanders and persister CAR-T cels, respectively, while $\kappa$ is the expansion rate of expanders, $\rho$ is the tumor growth rate and $\gamma$ is the CAR-T cell cytotoxicity rate (Fig 1). However, such analysis does not elucidate how the remaining four characteristic shape features of therapy response, namely, the CAR-T cell minimum after distribution, the CAR-T cell peak concentration ($C_{max}$), the CAR-T cell persistence level, and the minimum tumor burden achieved by therapy ($T_{min}$), correlate with model parameters, specially with the four remaining patient-specific parameters $A, B, \eta, \epsilon$. Here, $A$ is related to the antigen density of tumor cells, $B$ is related receptor density of CAR-T cells, $\eta$ is the activation rate of injected CAR-T cells, and $\epsilon$ is the rate of memory formation (transition from expander to persister phenotype, Fig 1). To address this question, we conducted a qualitative sensitivity analysis. Technically, we assess the effect of each of these 10 patient-specific parameters ($\mu_I, \kappa, \mu_E, \mu_P, \gamma, \rho, A, B, \eta, \epsilon$), one at a time, on these response hallmarks (Figs 4, S5 and S6). The two remaining model parameters, namely the tumor carrying capacity $K$ and the recruitment rate of persister CAR-T cells $\theta$, had no effect on these response hallmarks and were held constant across all patients (S5 Fig).

First, we found that the CAR-T cell minimum after the distribution phase is intrinsically related to the activation rate $\eta$, allowing us to obtain a first estimate for this parameter from the duration of the distribution phase, since the death rate of injected CAR-T cells is approximated by the distribution slope. We also identified that $C_{max}$ and $T_{min}$ are influenced by six parameters, $\kappa, \rho, \gamma, \mu_E, A$ and $B$, with higher $C_{max}$ associated with lower $T_{min}$. However, since that $\kappa, \rho, \gamma$ and $\mu_E$ are initially mapped, and thus fixed, by the characteristic slopes, only $A$ and $B$ can be varied to match $C_{max}$ and $T_{min}$. In other words, the initial estimates for the pair $(A, B)$ can be estimated from the pair $(C_{max}, T_{min})$. From the biological point of view, this indicates that CAR-T cell expansion and tumor response are related to both product and patient-related characteristics, including the densities of antigens in tumor cells

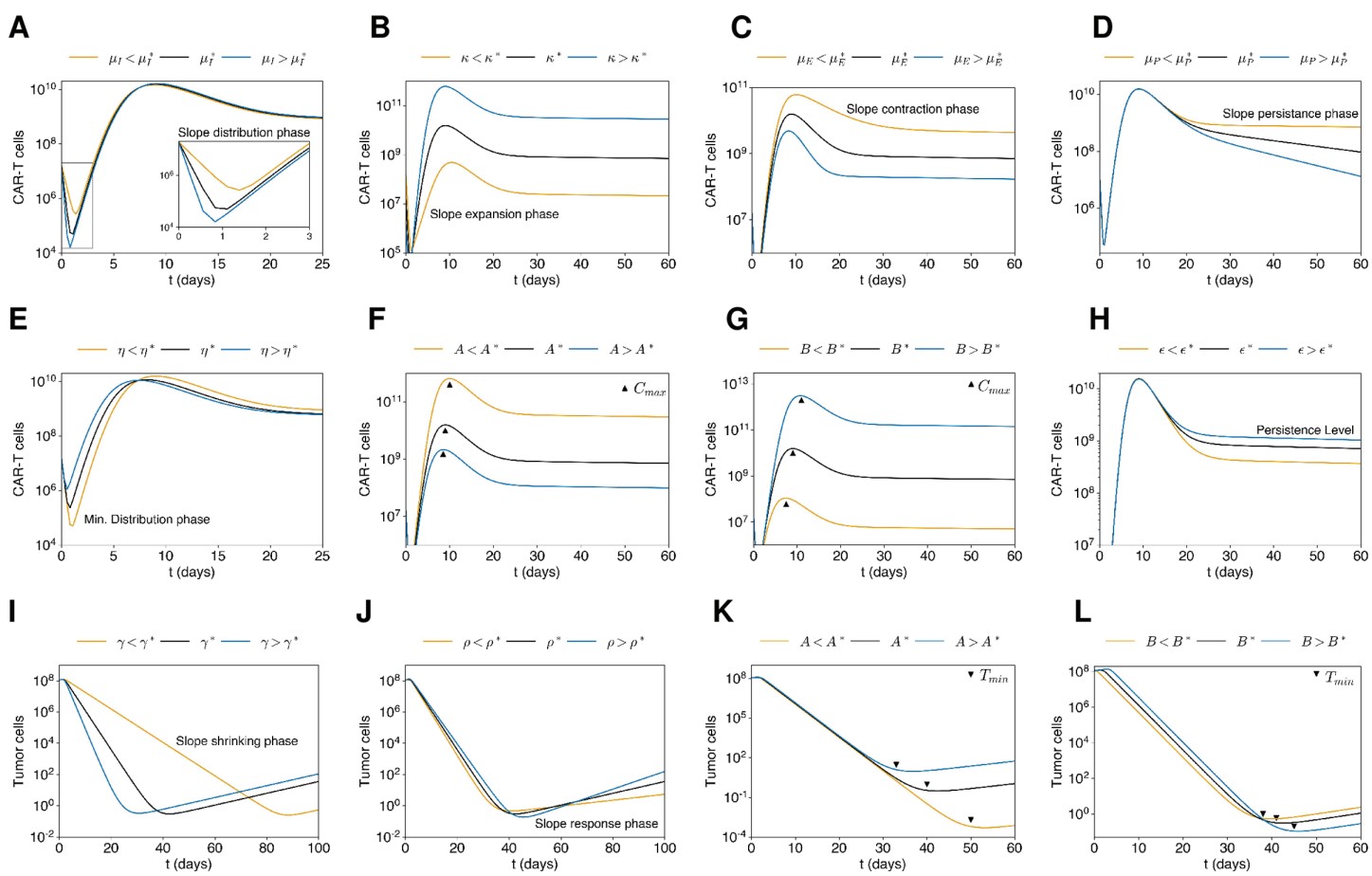

**Fig 4. Mapping of mechanistic parameters on response characteristics.** A qualitative sensitivity analysis identified a nonlinear mapping between ten different shape features of the model response (slopes $m_D, m_E, m_C, m_P, m_S, m_P$, CAR-T cell minimum, CAR-T cell peak concentration, minimum tumor load, and persistence level) to ten mechanistic parameters. This is illustrated by simulations for patient M44 (black) and alternative scenarios (blue and yellow) where one parameter is changed at a time. **A-D** The slopes in the distribution, expansion, and contraction phases are determined by the decay of injected CAR-T cells ($\mu_I$), the net expansion rate of CAR-T expander cells ($\kappa$ – $\mu_E$), their death and exhaustion ($\mu_E$), and the decay of CAR-T persister cells ($\mu_P$). **E** The minimum level of CAR-T cells observed at the end of the distribution phase is determined by the activation rate of injected CAR-T cells ($\eta$). **F,G** The CAR-T cell peak depends on the saturation constants for the antigen binding ($A$) and tumor killing ($B$) functions. **H** The persistence level is mainly determined by the memory pool formation rate $\epsilon$. **I,J** The slopes in the shrinkage and response phases of tumor dynamics are determined by CAR-T cell cytotoxicity ($\gamma$) and tumor growth rate ($\rho$), respectively. **K,L** The minimum tumor load is negatively correlated with the CAR-T cell peak and therefore is also determined by $A$ and $B$. Although most of the response characteristics appear to have a first-order dependence on only one mechanistic parameter, this is not the case for the CAR-T cell peak and the minimum tumor burden, which also strongly depend on $\kappa, \mu_E, \gamma$ and $\rho$. See also S5 and S6 Figs.

($A$) and receptors in CAR-T cells ($B$). Finally, we observed that the CAR-T cell persistence level is closely linked to the memory formation rate $\epsilon$ and on the CAR-T cell peak itself (along with the parameters governing $C_{max}$). Again, considering that the previous parameters were already mapped, this allows the estimation of $\epsilon$.

Therefore, this analysis identifies a mapping between the ten different shape features of the model response (phenomenological parameters, consisting of six characteristic slopes, the minimum tumor load, CAR-T cell minimum, peak concentration and persistence level) to ten mechanistic parameters ($\mu_I, \kappa, \mu_E, \mu_P, \gamma, \rho, \eta, \epsilon, A$ and $B$), and provides a rationale for estimating these parameters. This procedure (see Methods) utilizes the clinical time courses

to derive initial estimates for these 10 parameters and then employs an optimization algorithm to obtain optimal model fits for each patient (Fig 2). The efficacy of this strategy underscores the intimate relationship between each shape feature of the patient time courses and the model parameters, i.e., the mapping between the phenomenological parameters contained in the patient data and the mechanistic model parameters, which ensures identifiability. Thus, our model can be regarded as a minimal model for delineating multiphasic responses, where a biologically informed parameter estimation approach enhances the interpretability of parameter values and ensures that they are inherently linked to the dynamics described by the patient's time course.

In conclusion, we demonstrated how distinct model parameters dictate the shape of CAR-T and tumor cell response, providing a systematic approach for parameter estimation and revealing minimal redundancy in the model parameters.

## Clinical data suggests poor predictive value of patient-specific factors on IL-6 fold change, but strong temporal connection

So far we illustrated how each feature of the dynamical therapy response corresponds to distinct functional parameters of our model. However, it is well established that certain pharmacological and clinical factors such as CAR-T cell dose, CAR-T cell peak and its time, and initial tumor burden are associated with therapy response or can serve as indicators of severe side-effects [34–36]. Our approach allows us to obtain these and other quantitative estimates from the model fits and systematically correlate them with predicted clinical outcomes. Therefore, for each patient's fitted time course, we evaluated and correlated key quantities, including the fold change in IL-6 concentration from baseline to peak, the fold change for CAR-T cell counts from dose to peak, peak times for CAR-T cells, IL-6, and macrophages, as well as the tumor burden at days 28 and 90 (Fig 5).

Our analysis revealed that, for the given cohort of 15 patients, the simulated fold change in IL-6 concentration cannot be correlated with or predicted from the individual patient level based on clinical factors such as total CAR-T cell dose, initial tumor burden, CAR-T cell fold change, or time of IL-6 peak (Fig 5A–5D). This highlights that, at the patient level, the increase in IL-6 levels and its likely association with CRS does not depend solely on any of the above factors alone but appears to emerge from an interplay of various patient-specific factors.

Although no single marker for IL-6 fold change was found, our analysis confirmed the strong temporal connection between events after CAR-T cell infusion (Fig 5E–5F). The estimated CAR-T cell peaks occur within 0 to 5 days after the IL-6 peak for almost all patients, aligning with clinical observations [8]. Additionally, the cytokine peak was typically preceded by a peak of activated macrophages occurring 0 to 2 days earlier for all recorded patients. This suggests that IL-6 peak, and probably CRS, could be better anticipated by monitoring levels of activated macrophages. Although these findings need clinical validation they open a window of opportunity to intervene before a critical cytokine peak builds up, thereby lowering the risk of CRS.

We also investigated the remission level at days 28 and 90 post-therapy initiation and assessed their correlation with CAR-T cell expansion (Fig 5G–5I). We observe a very weak negative correlation between CAR-T cell fold change from dose to peak and the simulated residual tumor burden at day 28 ($R^2$=0.17), which which diminishes further by day 90 ($R^2$=0.05). Nevertheless, the direct correlation between responses at days 28 and 90 reveals a moderate association ($R^2$=0.47), indicating that early remission levels can partially predict later outcomes.

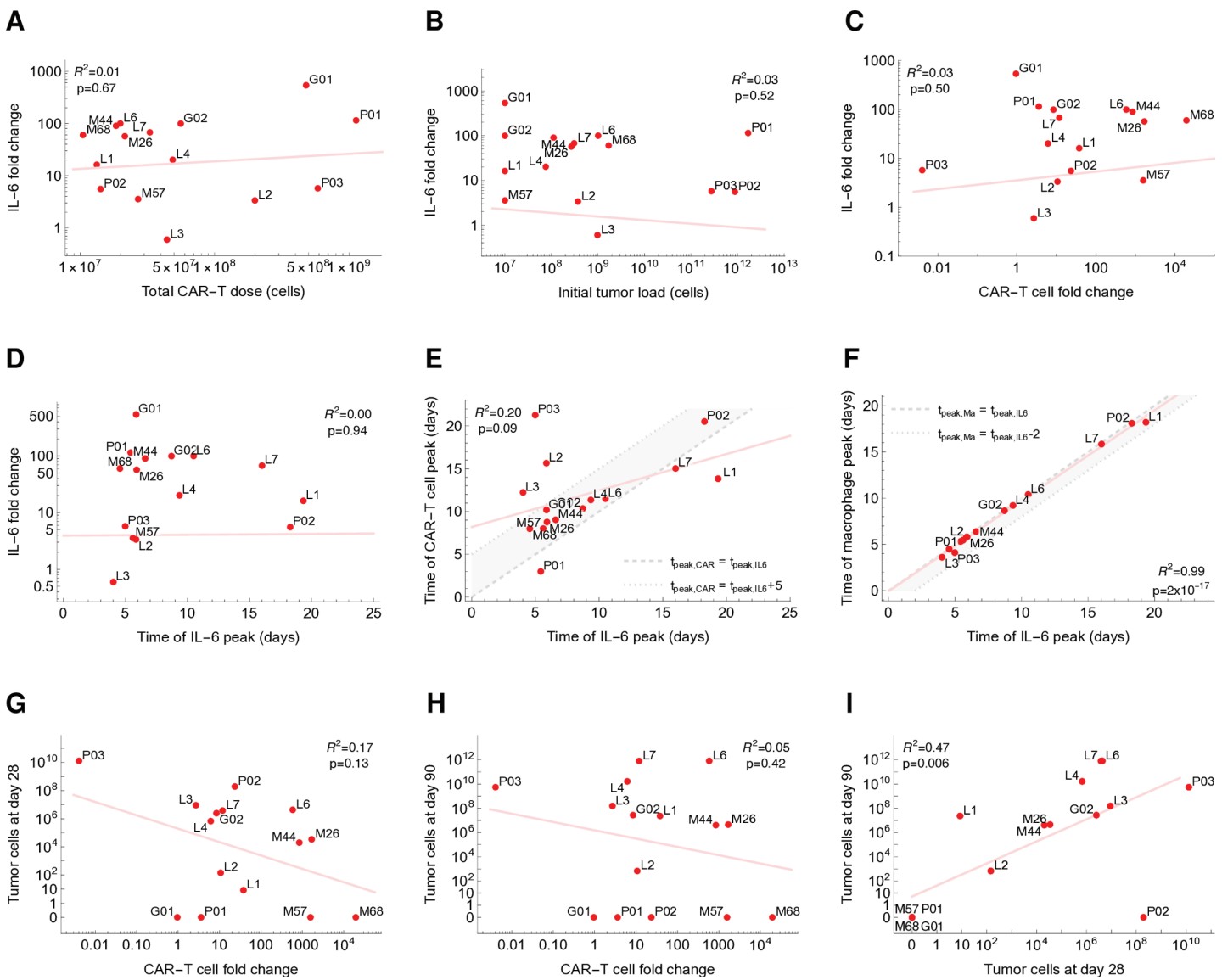

**Fig 5. Patient-specific pharmacological and clinical parameters evaluated from model fits. A–D** Within the patient level, the fold change in IL-6 concentration (baseline to peak) cannot be predicted from a single indicator, such as total CAR-T dose, initial tumor burden, CAR-T cell fold change (dose to peak), or time of IL-6 peak. **E** However, for the majority of patients, the model predicts that the peak of CAR-T cells occurs between 0 and 5 days after the cytokine peak (gray area). **F** For all patients, the cytokine peak is predicted to occur between 0 and 2 days after the peak of activated macrophages (gray area). **G,H** The tumor burden at days 28 and 90 post-therapy initiation shows a weak negative correlation with CAR-T cell fold change (a value of 0.1 cells is assigned for cases where the model predicts zero tumor cells). **I** The model predicts a moderate correlation between tumor burden responses at days 28 and 90. In each panel, the best linear fit for the red points is shown in light red, and the corresponding correlation coefficient $R^2$ and p-value are given.

In summary, while no single marker at the patient level reliably predicts the IL-6 peak and the likely associated occurrence of CRS, monitoring macrophage levels enables prediction of cytokine kinetics, thereby suggesting a potential clinical intervention.

## CAR-T cell dose and initial tumor burden influence individual treatment outcome

We showed that two crucial and clinical available parameters, namely CAR-T cell dose and initial tumor burden are insufficient to individually predict the severity of potential side

effects, reflected by the poor correlation with IL-6 fold change. However, in terms of our mathematical model we can evaluate how changes of these parameters effect treatment outcome on a cohort level. Using the optimal parameter set identified for each patient and changing only the CAR-T cell dose or initial tumor burden, we examine how a 10 and 100 fold change in these parameters affect the simulated response parameters, including CAR-T cell peak ($C_{max}$), remission levels ($T_{min}$) and long-term memory formation at days 28 and 90, IL-6 fold change (baseline to peak), and the time of CAR-T cell and IL-6 peaks (Figs 6 and S7).

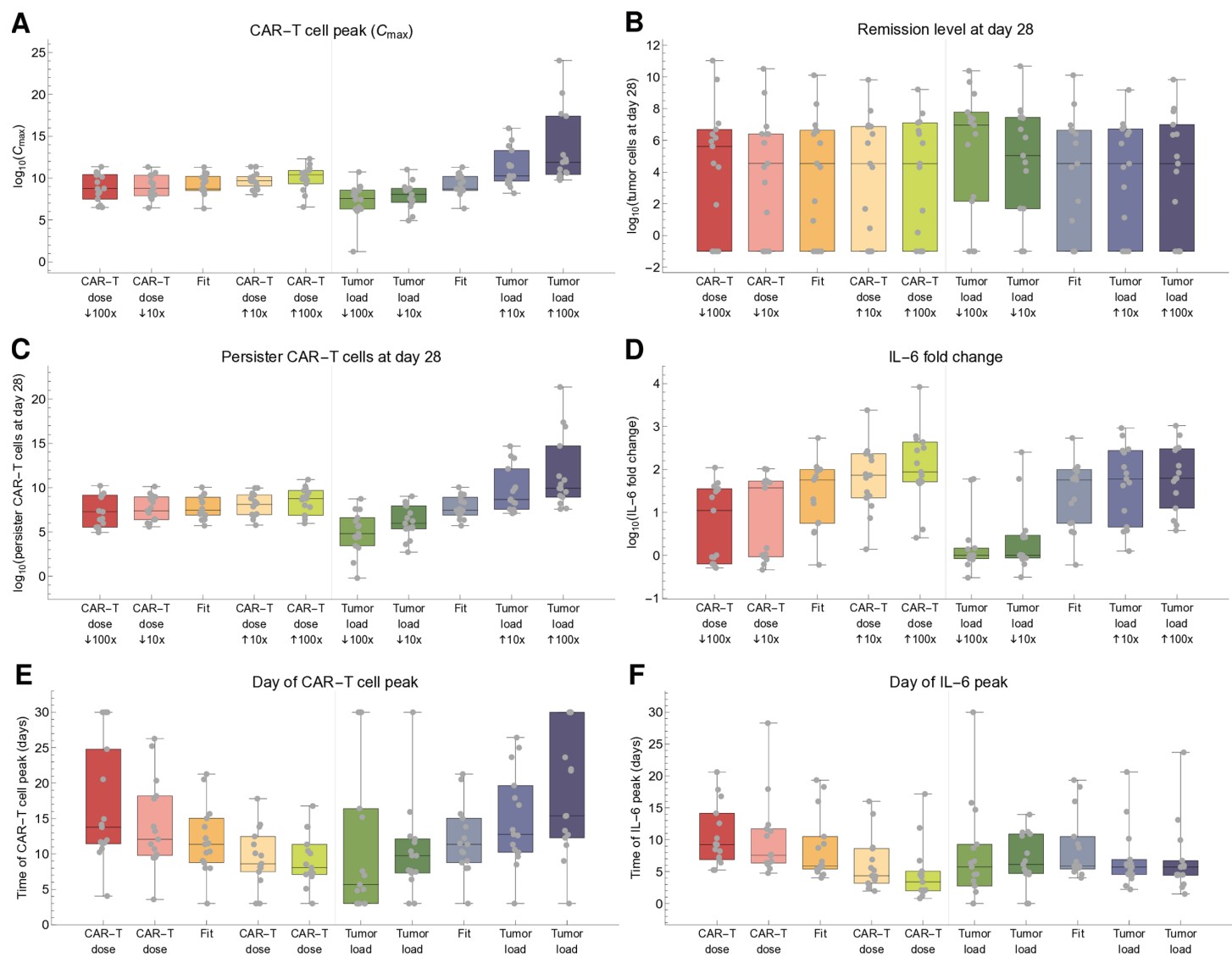

**Fig 6. Effect of simulating different dosing protocols on CAR-T cell dynamics, tumor response, and cytokine peak.** To investigate the outcomes of a different CAR-T dose or change in the preconditioning or bridging therapy, we compared the standard scenario (Fit) with simulations starting with either a different CAR-T dose or initial tumor burden (10x and 100x smaller and higher). Assessed outcomes were: **A** number of CAR-T cells at peak ($C_{max}$), **B** number of tumor cells at day 28, **C** number of persister CAR-T cells at day 28 **D** IL-6 fold change (baseline to peak), **E** day of CAR-T cell peak, **F** day of IL-6 peak. The number of tumor and CAR-T persister cells at day 90 were also assessed and did not present sensible differences (S7 Fig).

Upon simulating different CAR-T cell doses, we observed that the CAR-T peak concentration, remission levels and long-term memory formation at days 28 and 90 remained almost unaffected (Fig 6A–6C). However, smaller simulated doses resulted in decreased IL-6 fold change and delayed peaks of both CAR-T cells and IL-6 (Fig 6D–6F). Based on these model predictions, we reason that smaller CAR-T cell doses could extend the expansion phase without a proportional change in absolute cell numbers, thereby maintaining overall remission levels. This extended CAR-T cell expansion phase may lead to a more distributed activation of macrophages over time, as reflected by the delayed IL-6 peak and the reduced fold change from baseline to peak, potentially lowering the risk of severe CRS.

Initial tumor burden is an important factor that can be targeted during bridging and preconditioning therapy. We observed that reducing the initial tumor burden led to lower CAR-T cell peaks and IL-6 fold change (Fig 6A, 6D), with an earlier CAR-T cell peak and no substantial changes in the IL-6 peak timing (Fig 6E–6F). Additionally, modifying the initial tumor burden had minimal impact on remission levels at days 28 and 90, but higher tumor burdens were associated with increased long-term memory formation (Fig 6B–6C).

Although we see a correlation between IL-6 peak levels and both the CAR-T cell dose and the initial tumor burden (Fig 6D), this relationship does not appear to be directly proportional. Specifically, when the initial simulated tumor burden increases 10x and 100x, the IL-6 peak levels continue to rise less proportionally, suggesting a saturation effect. A similar but less pronounced trend is observed for variations in CAR-T cell dose. We interpret this saturation as a result of the functional model setup in which antigen-binding and DAMPs act as a triggers rather than the primary mechanisms for macrophage activation (see below).

In conclusion, our findings suggest that adjusting the administered CAR-T cell dose may exert a greater influence on kinetic parameters such as peak times and IL-6 peak concentration. The initial tumor burden and the corresponding antigen load appears to serve as a catalyst for CAR-T cell expansion, potentially explaining higher persistence of CAR-T cells but also increased severity of cytokine-related side effects.

## Disentangling mechanisms of macrophage activation and IL-6 release

While it appears that cytokine-related side effects can be reduced by adjusting CAR-T cell dose or initial tumor burden, targeting macrophage-mediated cytokine release directly is also a clinical option. A quantitative understanding of the temporal dynamics of macrophage activation and IL-6 release is crucial for this purpose.

Our model assumes three distinct molecular and cellular mechanisms for macrophage activation, namely DAMPs release, antigen-binding, and CD40 contact, described by parameters $\beta_K$, $\beta_B$ and $\beta_C$ respectively (Eq (11)). To unveil the dynamics underlying these mechanisms, we simulated the time course of patient M44, selectively activating one mechanism at a time (Fig 7A). Activation via DAMPs release alone ($\beta_K > 0$, $\beta_B = 0$, $\beta_C = 0$) led to IL-6 peak around day 3, during the tumor shrinkage phase. Activation solely through antigen recognition ($\beta_B > 0$, $\beta_K = 0$, $\beta_C = 0$) resulted in IL-6 concentration rising concurrently with CAR-T cell expansion, peaking around day 8. Considering only CD40-mediated activation ($\beta_C > 0$, $\beta_K = 0$, $\beta_B = 0$), IL-6 kinetics remained at baseline levels due to the requirement of previous infiltration by naive macrophages. Yet, integrating CD40-mediated activation with small values for DAMPs release and antigen-binding parameters ($\beta_K \gg 0$, $\beta_B \approx 0$, $\beta_C \approx 0$), the IL-6 peak occurred around day 13, four days post-CAR-T cell peak (purple curve). These findings suggest that each mechanism occurs at distinct time points, influenced by various interactions unfolding after infusion: DAMPs drive macrophage activation during the tumor shrinkage

phase, followed by antigen-binding-mediated activation during CAR-T cell expansion, and contact-dependent CD40-mediated activation.

To understand the individual contributions of each mechanism, we divided the activated macrophages into three sub-populations based on their source (DAMPs-activated, antigen-activated, CD40-activated) and, for each patient, determined the peak time and cumulative number of activated macrophages for each sub-population (see Methods), and the CAR-T cells and IL-6 peak times (Fig 7B). Interestingly, the previously described timeline holds true

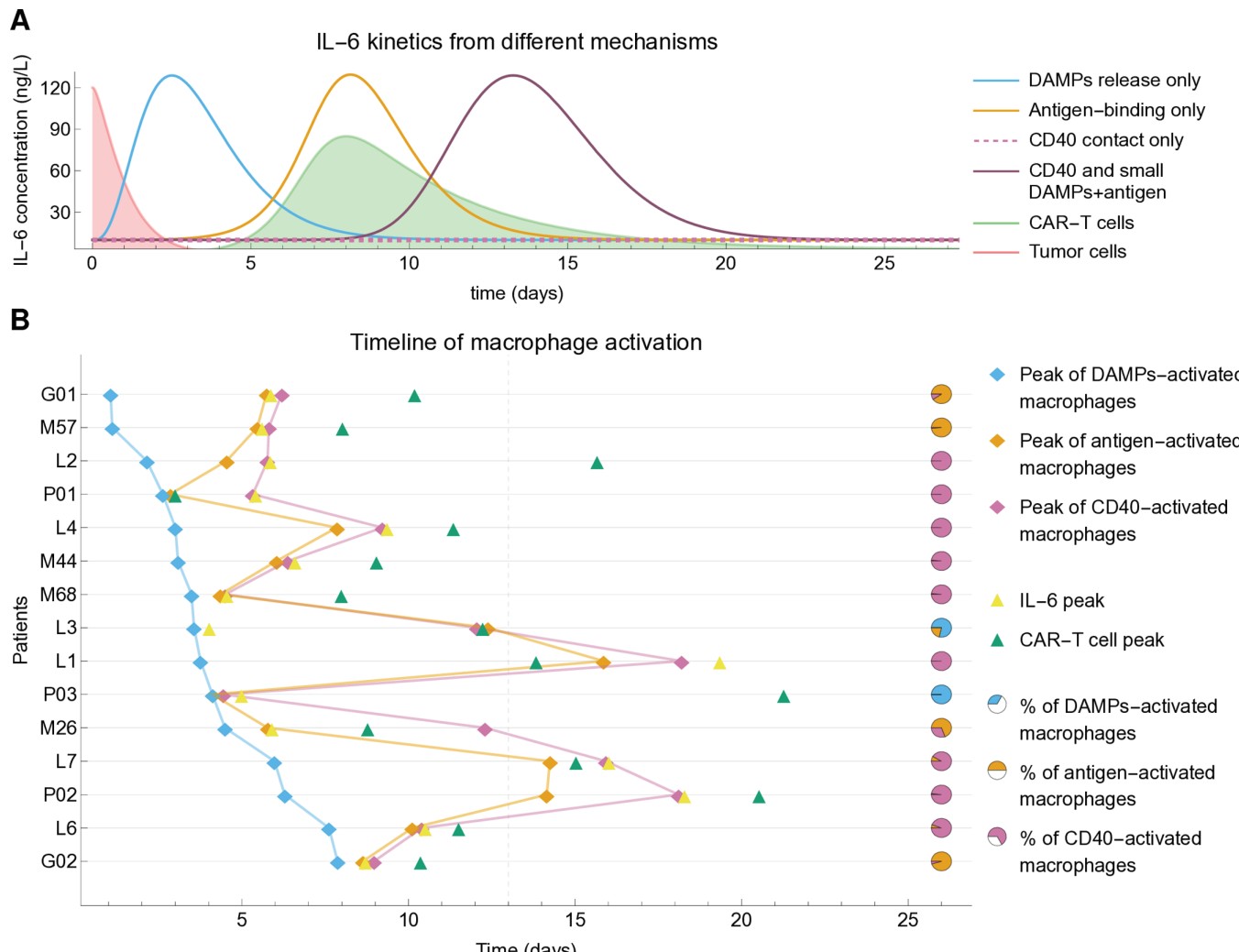

**Fig 7. Dynamics of macrophage activation and IL-6 release. A** Time course of patient M44, selectively activating DAMPs release, antigen-binding, and CD40 contact mechanisms one at a time. DAMPs-only activation leads to an IL-6 peak around day 3 during tumor shrinkage. Antigen-biding activation results in IL-6 increasing with CAR-T expansion, peaking around day 8. CD40-mediated activation alone maintains IL-6 at baseline, but when combined with small values for DAMPs and antigen-binding parameters, it produces an IL-6 peak around day 13, four days after the CAR-T cell peak. **B** Splitting the active macrophage population according to their source, i.e., activation mechanisms (see Methods), and reconstructing the timeline for each patient, we found that the peak of DAMP-activated macrophages takes place early, during the tumor shrinkage phase. The second subpopulation to present a peak is the antigen-binding-activated macrophages, which expand in parallel with CAR-T expansion. The last sub-population to peak are CD40-activated macrophages, which require the presence of previously activated macrophages. For 8 out of the 15 patients, the major source of activated macrophages was CD40 contact (pie charts), which was also the source of 62% of all activated macrophages when all patients are combined.

for the majority of patients: DAMP-activated macrophages reach their peak levels first, followed by the peak of antigen-activated macrophages, followed by the peak of CD40-activated macrophages.

While all these mechanisms can in principle equally contribute to macrophage activation, a computational estimate of the cumulative number of activated macrophages in each sub-population revealed that the majority of activated macrophages (62% considering all patients) are activated by the CD40 contact-dependent mechanism (pie charts in Fig 7B). This intriguing discovery suggests that macrophage activation mediated by contact-independent mechanisms, such as DAMPs release and the release of GM-CSF and IFN-$\gamma$ by antigen-binding-activated CAR-T cells, is limited and may not be the primary source of cytokine release. On the contrary, our findings propose that once CD40-mediated activation is triggered by these previous mechanisms, it enters a feedback loop in which activated macrophages in contact with CAR-T cells further activate more macrophages, leading to massive cytokine release. These results confirm that CD40–CD40L interactions are not essential for eliciting cytokine release but are likely the primary drivers exacerbating macrophage activation and IL-6 production, potentially increasing CRS severity. To address the robustness of these results, an extensive analysis comparing the full model with its reduced versions considering one or two mechanisms only confirmed our findings (Methods, S8 and S9 Figs ).

In summary, our findings indicate that macrophage activation via DAMPs release, antigen-binding, and CD40 contact unfolds sequentially during therapy, with CD40-mediated inflammation playing a predominant role.

## Targeted interventions to control macrophage activation

The deconvolution of the different mechanisms driving macrophage activation hints towards a clinical intervention to mitigate CRS. Currently, the primary drug used to address CRS is Tocilizumab, a monoclonal antibody that acts as an IL-6 antagonist [12]. However, targeting macrophage activation with GS-CSF and CD40 antibodies has also been explored in cell-based studies [37].

To mimic the use of such antibodies that specifically inhibit macrophage activation, we applied our patient-specific model but reduced the parameters related to macrophage activation, $\beta_K$, $\beta_B$ and $\beta_C$ one at time (Fig 8A). We observe that the IL-6 peak decreased with increased blocking, with varying effects depending on the patient. Overall, for a 50% reduction in the respective activation parameter, the mean reduction in the IL-6 peak was 7%, 12%, and 34% for DAMP, antigen-binding, and CD40, respectively. Repeating the same simulations for reduced, alternative models showed a similar or greater reduction in the IL6 peak when CD40 activation is reduced by 50% (S9 Fig). In addition, sensitivity analysis showed that these values did not change substantially when each of the eight fitted parameters was varied by 50% (S10 Fig). This suggests that the CD40-CD40L interaction is not only a major driver of cytokine release but also a promising target for clinical interventions.

To demonstrate the clinical potential of CD40 blocking, we conducted a series of model simulations reducing, for each patient, the CD40 parameter to 50% of the original value starting at different time points after infusion and keeping the block until either before, at or after the CAR-T cell peak (Fig 8B). We found that, on average, initiating the blocking between 0 (immediately after infusion) and 3 days after CAR-T cell infusion is most effective, with the block being maintained until or after the CAR-T cell peak. On average, a 30-40% reduction in IL-6 peak can be achieved with some patients showing even stronger responses.

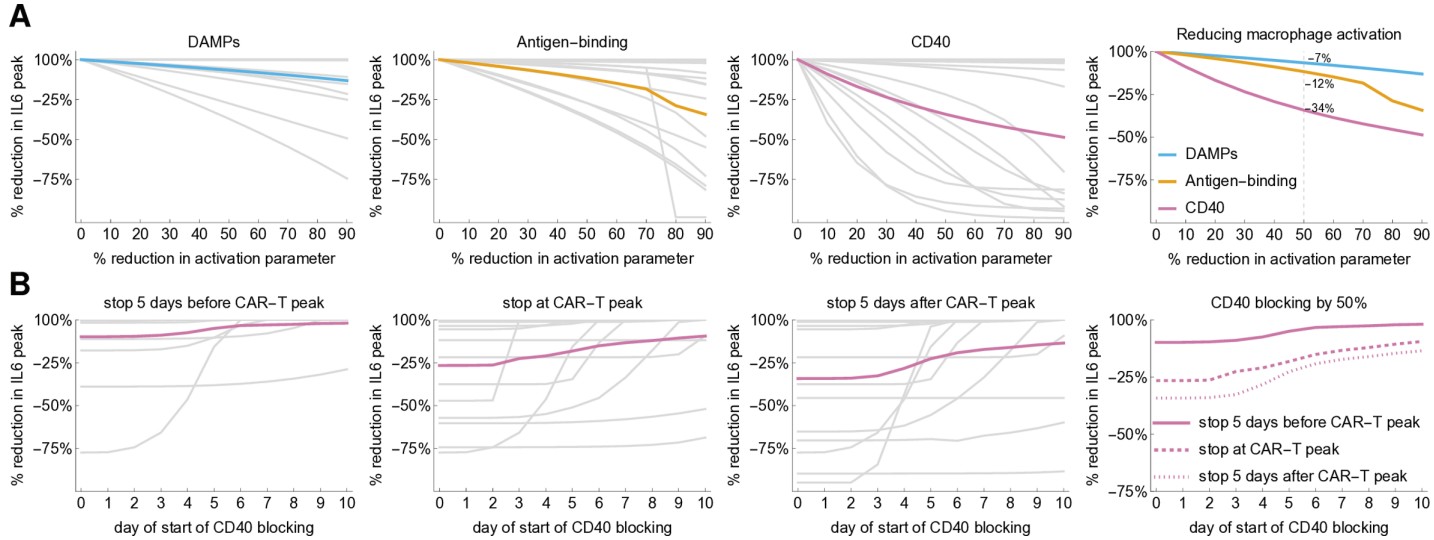

**Fig 8. Blocking of macrophage activation and cytokine release. A** Simulation results of interventions reducing mechanisms of macrophage activation one at a time, showing the reduction in IL-6 peak for each patient (gray curves) and mean effect (colored curves). For a simulated 50% reduction (dashed line, fourth panel) the overall reduction in IL-6 peak is indicated for each mechanism. **B** Simulation results of interventions blocking CD40 starting 0-10 days after CAR-T cell infusion, stopping -5, 0 and 5 days after CAR-T cell peak. The plots show the resulting reduction in IL-6 peak for each patient (gray curves) and the mean effect (colored curves).

Our model of sequential macrophage activation suggests that targeting the CD40-CD40L axis is an effective and clinically feasible strategy to control macrophage activation and reduce IL-6 peak height.

## Discussion

We developed a novel multi-layer mathematical model to investigate tumor response and CRS following CAR-T cell immunotherapy. The model accurately describes the timeline of therapy response and relates its dynamic shape to interpretable model parameters. Our findings highlight the role of early macrophage activation in cytokine release, thereby outlining potential strategies for clinical monitoring and prevention of CRS. By disentangling different modes of macrophage activation we identified the CD40-CD40L axis as a clinically feasible target to control the activation process and modulate IL-6 peak height.

The diverse configuration and clinical outcomes of patient responses to CAR-T cell therapy is attributed to product- and patient-specific factors [34]. Our model unveiled the mechanisms underlying multiphasic therapy response, showing how each of its shape features is linked to a specific model parameter. While, on one hand, this close correlation is the basis for a model-informed stepwise parameter estimation strategy, it, on the other hand, explains how key clinical indicators (CAR-T cell peak, persistence, and minimal tumor burden) are modulated by product-related factors (CAR-T cell expansion, cytotoxicity and receptor density) and patient-specific variables (tumor growth rate and antigen density).

In our analysis of patient cohorts, neither the initial CAR-T cell dose nor the baseline tumor burden proved to be reliable predictors for treatment outcomes. These factors, along with maximum CAR-T cell concentration, CD4/CD8 ratio, pre-infusion phenotypes, area under the concentration-time curve (AUC), have been extensively studied but lack consensus as prognostic factors [27,31,38–40]. On the other hand, we found that the minimal tumor

burden is inversely related to the CAR-T cell peak concentration, consistent with [41], which reported that higher peaks of CAR-T cells within 15 days are associated with increased likelihood of achieving complete remission.

While the precise determinants of CRS severity remain debated, some studies found tumor burden, lymphodepletion, CAR-T cell dose, and CAR-T product are independent predictors of CRS [42,43] and others suggested that initial tumor burden and CAR-T cell dose correlate with the severity of CRS [44–46]. In our patient-level analysis, neither CAR-T cell dose nor initial tumor burden reliably predicted IL-6 increase (as a surrogate for CRS severity) a priori. However, our cohort simulations shows that smaller CAR-T cell doses reduced IL-6 fold change, potentially lowering severe CRS risk without sensitive affecting CAR-T cell peak, remission levels, or memory formation. These findings aligns with the previously reported association between the expansion of CAR-T cells (peak/dose) and severity of CRS [38]. Also, a lower initial tumor burden reduced IL-6 levels but lowered both CAR-T cell peak and persistence, with minimal impact on remission levels. In [47], markers related to tumor burden and inflammation, both of which may be influenced by the underlying tumor biology, were highly associated with clinical outcomes.

Several mathematical models have been developed to understand T cell and cytokine interactions in different contexts [48–51]. For CAR-T cell therapy, Mostolizadeh et.al [23] developed a model that includes CAR-T cells, healthy and cancer cells, and cytokines, and applied optimal control theory for controlling cytokine release syndrome with tocilizumab. Hardiansyah and Ng [22] implemented a quantitative systems pharmacology model that links CAR-T cell kinetics to cytokine release, showing a correlation between disease burden and CAR-T cell expansion. Zhang et al. [25] built a computational model of CRS during CAR-T cell therapy, illustrating how cellular and molecular interactions among CAR-T cells, B-ALL cells, bystander monocytes, and various inflammatory cytokines affect CRS severity. In [24], the modulation of drug administration frequency and duration to maintain efficacy while preventing cytokine storms was investigated. While some of these models have been validated with clinical data, none explicitly include how different macrophage-associated mechanisms influence the extent and temporal dynamics of CRS.

Mouse models showed that CRS involves a multicellular network comprising CAR-T and host cells, with macrophages playing a crucial role [14,16]. A recent review [8] proposes that macrophage-mediated CRS emerges from antigen-binding-mediated inflammation, DAMPs release and CD40 contact. Built upon these concepts, our model accurately replicates these dynamics, showing an early peak of activated macrophages followed by cytokine peak within 1–2 days, then a CAR-T cell peak within 1–5 days. Disentangling these mechanisms, we showed that DAMPs underlie early cytokine release, while antigen-binding drives cytokine increase parallel to CAR-T cell expansion, and CD40 contact leads to late cytokine rise. A deeper analysis indicates that CD40 is the latest but the main driver of macrophage activation, suggesting that antigen-binding and DAMPs act more as triggers than primary mechanisms.

The preponderant role of CD40-contact in macrophage activation poses a candidate for interventions against CRS. Simulating a reduction of the different activation mechanism showed that targeting CD40 leads to the most substantial reduction in IL-6 peak height, causing an average reduction of 34% in the case of constant blocking and 33% reduction when blocking from 0-3 days after infusion to 5 days after CAR-T cell peak. This aligns with a recent cell-based in vitro model for CRS that showed that IL-6 supernatant levels decreased 20-30% when blocking cytokine release by bystander macrophages with neutralizing antibodies for GM-CSF and CD40, and decreased 30–40% due to the genetic disruption of the CD40L and/or CSF2 knock-out CAR-T cells [37]. Analyzing 54 relapsed or refractory MM patients treated with anti-CD19 and anti-B-cell maturation antigen CAR-T cells, Wang et al

[52] reported a median time of 2 days (range 0 to 5) to CRS onset for the severe CRS group (7 patients), whereas the mild CRS group (47 patients) had a median onset of 7 days (range 0 to 13). In our view, this suggests that additional pathways (e.g., DAMPs or other mechanisms) may be involved as major drivers in those rare cases of early and severe CRS.

Our model is based on a simplified yet widely accepted concept of CAR-T cell development, progressing from naive to expander to persister phenotypes. This framework is consistent with established views of T cell biology [53,54] and builds on recent mechanistic models of CAR-T cell dynamics [29,31]. To keep the model parsimonious, we assume that CAR-T expanders represent a composite of effector and expanding memory cells, while persisters comprise long-lived memory cells. The inclusion of a bidirectional, antigen-dependent phenotypic switch introduces plasticity without requiring additional differentiation pathways. This approach reduces model complexity, ensuring that each parameter corresponds directly to a specific feature of multiphasic CAR-T and tumor cell dynamics. While alternative differentiation pathways and mixed phenotypes were not included, these factors are unlikely to significantly affect the robustness of our conclusions. However, further research is needed to clarify how specific differentiation pathways influence CAR-T cell treatments.

The nature of our research was investigative rather than predictive, aiming to understand the mechanisms of therapy response and macrophage-mediated CRS rather than making predictions at a populational level. While our model provides insights into these processes, it also faces limitations. A notable strength of our approach is the one-to-one correspondence between parameters and time course shape features, which enables a comprehensive understanding of system dynamics and partially mitigates the challenge of parameter identifiability. However, this process relies heavily on the availability of tumor data at multiple time points, which is often lacking in clinical studies [1,32,55–57]. While collecting time-resolved tumor cell data poses clinical and technical challenges, particularly for certain tumor entities, it represents an essential step toward more robust and clinically applicable modeling frameworks and will be instrumental in realizing the full potential of predictive mathematical models to advance the field.

Although preclinical models indicated that IL-6 plays a role in macrophage activation, the exact degree of this contribution remains uncertain [14]. Due to the need for additional data to accurately calibrate this relationship, we opted not to incorporate a feedback mechanism of IL-6 on macrophage activation, regarding it as a second-order effect. In addition, the use of macrophage time courses may improve our fitting strategy, which estimates IL-6 and macrophage parameters based on IL-6 data only.

Finally, while we use IL-6 fold increase to indicate CRS severity, there is no standardization in clinical grading scales [58], which are often based on symptoms or therapeutic interventions rather than measurable thresholds [59–61]. Increasing evidence showed that IL-6 plays a pivotal role in CRS, and elevated IL-6 levels are consistently associated with severe CRS in clinical studies [8,62]. In particular, tocilizumab, an IL-6 receptor antagonist, has become a first-line treatment for CRS, further underscoring its relevance [63]. However, we acknowledge that CRS severity is a complex, multifactorial process influenced by multiple cytokines (e.g., IL-1, TNF-$\alpha$, IFN-$\gamma$) and cellular interactions beyond IL-6 alone [8,62]. In this context, our model interprets IL-6 as a surrogate marker of macrophage-mediated CRS rather than a definitive predictor of its severity [64,65]. While further experimental validation is needed to establish direct correlations between IL-6 and other markers and severe CRS, our modular framework can be extended to include additional pathways.

Our study demonstrates the efficacy of a compact multi-module mathematical model in accurately describing a retrospective dataset from patients undergoing CAR-T cell therapy. Our findings shed light on the timeline and influence of macrophage activation on cytokine release and pinpoint the CD40-mediated mechanism as the primary driver of cytokine release, suggesting it as a potential target for CRS treatment strategies.

## Methods

### Clinical data

We did an extensive search in the literature for datasets containing patient time courses with enough data points for fitting multiphasic CAR-T kinetics and IL 6 kinetics. We then included in our modeling all datasets satisfying these requirements. Individual pharmacokinetic profiles of patients undergoing anti-CD19 CAR-T cell therapy were digitized from [1,32,55,56], and we collected the raw data disclosed in [57]. A detection limit was set for CAR-T at 25 copies/$\mu$g DNA [57] or $2.5\text{x}10^5$ cells (for conversions see Model Setup) and relapse was defined as the absolute number of total tumor cells exceeding its initial tumor burden.

We incorporated data from [1] for two pediatric patients. Patient 1 received a total of $1.2\times10^7$/kg CAR-T cells administered over three consecutive days, while Patient 2 received a single infusion of $1.4\times10^6$/kg CAR-T cells. Although specific initial tumor burden data were not provided, CD19 expression analysis in bone marrow samples of Patient 2 revealed a pretreatment blast population comprising approximately 7.7% CD19- cells and 92.3% CD19+ cells. In the absence of peripheral blood measurements, we assumed the same distribution of antigen-negative and positive cells to set the initial tumor burden. Patient 2 experienced a clinical relapse evident in the peripheral blood two months post-infusion, while Patient 1 achieved clinical remission that persisted for nine months.

The dataset from [32] includes a phase I study with pediatric B-ALL patients treated using a second-generation CAR with a 4-1BB intracellular co-stimulatory domain. Each patient received one CAR-T cell infusion, with initial tumor burden estimated from pretherapy peripheral blood blast percentages. Progression-free survival (PFS) days was used as relapse times. The IL-6-profile was represented as a heatmap for the first month, and we computed the average value to capture the time series results of IL-6 concentrations in serum.

We used data from [55], patients 1, 2, 3, and 4 diagnosed with ALL received treatment using autologous CAR with CD28 intracellular co-stimulatory domain, while patients 6 and 7 were treated with autologous CAR featuring CD137 intracellular co-stimulatory domain. The time-profile of CD19+ cells in lymphocytes in the blood, measured by flow cytometry, revealed that within 2-8 months, patients experienced either progression or relapse of CD19+ leukemia cells.

Finally, we used data from [57], for adult chronic lymphocytic leukemia (CLL) patients treated with split-dose of autologous T-cells transduced with a CD19-directed CAR. With the exception of patient 22, who transformed to CD19-dim DLBCL, patients exhibited either complete or partial responses. For patients P01, P02 and P03, the initial CLL tumor burden and cytokine information (baseline and fold change) were extracted from [56] (respectively named UPN01, UPN03 and UPN02). Initial tumor burden on day -1 was available for UPN02 (P03), while for UPN01 (P01) and UPN03 (P02), only bone marrow data was provided, we assumed a similar approximation in blood due to its reasonable range.

An overview of patient and product characteristics and the corresponding infused doses and regimens are presented in S1 Table.

## Mathematical modeling

**Modeling CAR-T cell dynamics and tumor response.** Starting from our previous model [5], we developed a simplified version aimed at describing, with a minimal number of parameters, the multiphasic dynamics of CAR-T cells as a process emerging from the interaction between CAR-T phenotypes and their binding to antigen-expressing tumor cells (Fig 1). The model consists of four compartments representing three CAR-T cell phenotypes, namely injected, expanders, and persisters, represented as $C_I(t)$, $C_E(t)$, and $C_P(t)$, respectively, and a population of antigen-positive tumor cells, represented as $T_P(t)$. The model is given by the following system of ordinary differential equations (ODEs):

$$\frac{dC_I}{dt} = -\eta F(T_P)C_I - \mu_I C_I, \tag{1}$$

$$\frac{dC_E}{dt} = \eta F(T_P)C_I + \kappa F(T_P)C_E - \epsilon(1 - F(T_P))C_E + \theta F(T_P)C_P - \mu_E C_E, \tag{2}$$

$$\frac{dC_P}{dt} = \epsilon(1 - F(T_P))C_E - \theta F(T_P)C_P - \mu_P C_P, \tag{3}$$

$$\frac{dT_P}{dt} = \rho T_P\left(1 - \frac{T_P}{K}\right) - \gamma \frac{C_E}{B + C_E}T_P. \tag{4}$$

The function $F(T_P)$ represents the antigen-receptor biding interaction and is described by a Michaelis-Menten functional response

$$F(T_P) = \frac{T_P}{A + T_P}, \tag{5}$$

where $A$ is the half-saturation constant. Following the approach in [26], we assume that T cell response rates are based on their phenotype – injected cells become expanders, expander cells proliferate, and persister cells are reactivated – and proportional to $F(T_P)$, while the rate of persister cell formation is proportional to $1 - F(T_P)$. The decay rates for CAR-T cell phenotypes include natural death and distribution throughout the body for injected cells ($\mu_I$), natural death and exhaustion for expanders ($\mu_E$), a slower natural death rate for persisters ($\mu_P$). A logistic growth is assumed for tumor cells with rate $\rho$ and carrying capacity $K$. Following [27], the killing of tumor cells is described with a Holling type-II response with saturation for expanders cells, further simplifying our previous model [5]. An overview of model parameters is presented in S2 Table. Detailed information on model setup and parameter estimation is provided below.

**Modeling antigen-negative tumor relapse** The basic model is extended to include a compartment of antigen-negative tumor cells. Emerging from a simplification of a recent model for CAR-T cell therapy resistance [7], this second module has two ODEs describing the dynamics of antigen-positive ($T_P$) and antigen-negative ($T_N$) tumor cells, given by

$$\frac{dT_P}{dt} = \rho T_P\left(1 - \frac{T_P + T_N}{K}\right) - \gamma \frac{C_E}{B + C_E}T_P, \tag{6}$$

$$\frac{dT_N}{dt} = \rho T_N\left(1 - \frac{T_P + T_N}{K}\right) - g_0\gamma \frac{C_E}{B + C_E}T_N. \tag{7}$$

The killing of antigen-negative cells is multiplied by a fraction $g_0$, representing a reduced cytotoxicity due to antigen absence. A small but non-zero value for $g_0$ is assumed to account for a minimal cytotoxic effect due to the bystander effect and/or the CAR-T cell's endogenous TCR [7,33,66]. Antigen-negative tumor cells do not contribute to the antigen-dependent process of CAR-T cell activation, expansion, and memory recruitment. Thus, the model with antigen-positive and antigen-negative tumor cells is given by Eqs (1)–(3), (5) and (6)–(7).

**Modeling macrophage-mediated cytokine release**   A third layer is added to describe cytokine dynamics. In line with widely accepted biology [15,16,67], we assume that cytokine release is mediated by macrophage activation. This activation is driven by three mechanisms: i) upon target recognition, expander CAR-T cells release cytokines and soluble inflammatory mediators that activate macrophages, such as interferon-$\gamma$, granulocyte–macrophage colony-stimulating factor and tumor necrosis factor; ii) the killing of tumor cells releases DAMPs that further amplify macrophage activation; and iii) macrophage activation is also promoted by a contact-dependent mechanism, through the expression of CD40 and CD40-ligand by macrophages and CAR-T cells, respectively. We then assume that activated macrophages secrete inflammatory cytokines, including IL-6, chosen due to its significant role in CRS and the availability of patient time courses containing IL-6 kinetics alongside CAR-T cell responses. Under these assumptions, the model is

$$\frac{dM_i}{dt} = \sigma_M - h(C_E, T_P, M_a)M_i - \delta_M M_i, \tag{8}$$

$$\frac{dM_a}{dt} = h(C_E, T_P, M_a)M_i - \delta_M M_a, \tag{9}$$

$$\frac{dIL_6}{dt} = \sigma_I + \alpha M_a - \delta_I IL_6, \tag{10}$$

where $M_i$ and $M_a$ represent the naive (monocytes) and activated macrophages, respectively, and $IL_6$ describes the IL-6 concentration. Furthermore, $\sigma_M$ is the natural production rate of naive macrophages, $\delta_M$ is the death rate of macrophages, $\sigma_I$ represents the endogenous production rate of IL-6, $\delta_I$ is the IL-6 natural decay rate, and $\alpha$ is the rate of IL-6 release by activated macrophages. Due to the lack of macrophage time courses and to maintain parsimony, equal death rates were assumed for activated and inactivated macrophages.

The macrophage activation rate is given by

$$h(C_E, T_P, T_n, M_a) = \beta_B \frac{T_P}{A + T_P} C_E + \beta_K \frac{C_E}{B + C_E}(T_P + g_0 T_N) + \beta_C \frac{M_a}{C + M_a} C_E, \tag{11}$$

and encompasses the contribution of the three different mechanisms for macrophage activation: antigen-binding-mediated release of inflammatory signals by CAR-T cells (first term, proportional to CAR-T activation, with contribution $\beta_B$), tumor-killing-mediated release of DAMPs (second term, proportional to tumor killing, with contribution $\beta_K$), and CAR-T cell and macrophages contact (third term, Michaelis-Menten kinetics for CD40-CD40L binding, with contribution $\beta_C$).

**Approximating solutions and predicted slopes for multiphasic dynamics.**   Here, we present a rationale that allows to identify the main mechanism driving each phase of the CAR-T and tumor cell multiphasic dynamics. We simplify the ODEs for total CAR-T and

tumor cells at the different time scales and this leads to the calculation of approximate solutions for each phase, along with their characteristic slopes. These arguments are supported by a quantitative approach in which, for each patient, we calculated the characteristic slopes from the model solution at each phase and compared them with the predicted parameter governing each slope (Fig 3).

The total CAR-T cell population is given by $C_T = C_I + C_E + C_P$ and is described by

$$\frac{dC_T}{dt} = -\underbrace{\mu_I C_I}_{\text{Injected death}} + \underbrace{\kappa F(T_P) C_E}_{\text{Expander proliferation}} - \underbrace{\mu_E C_E}_{\text{Expander depletion}} - \underbrace{\mu_P C_P}_{\text{Persister death}} .$$

During the distribution phase, the CAR-T cell population consists mainly of injected cells, from which we approximate $C_T \approx C_I$ and $C_E, C_P \approx 0$. Thus, $dC_T/dt \approx -\mu_I C_T$, leading to an approximating solution $C_T = C_0 e^{-\mu_I t}$ with slope $-\mu_I$. During the expansion phase, the tumor burden is high and the CAR-T cell population consists mainly of CAR-T expanders, leading to the approximation $C_T \approx C_E$, $C_I, C_P \approx 0$, $F(T_P) \approx 1$. The approximating ODE is $dC_T/dt \approx (\kappa - \mu_E) C_T$, leading to an approximating solution $C_T = C_{\min} e^{(\kappa - \mu_E)t}$ with slope $\kappa - \mu_E$. Similarly, in the contraction phase the CAR-T cell population presents the same characteristics, but with a reduced tumor burden, leading to $F(T_P) \approx 0$ and $dC_T/dt \approx -\mu_E C_T$. The approximating solution reads as $C_T = C_{\max} e^{-\mu_E t}$ with slope $-\mu_E$. Finally, the persistence phase is characterized by the presence of CAR-T persisters, leading to $C_T \approx C_P$, $C_E, C_I \approx 0$ and an approximating solution $C_T = C_{\text{con}} e^{-\mu_P t}$ with slope $-\mu_P$.

Similarly, the total tumor population, given by $T = T_P + T_N$, is described by

$$\frac{dT}{dt} = \underbrace{\rho T \left(1 - \frac{T}{K}\right)}_{\text{Tumor growth}} - \underbrace{\gamma \frac{C_E}{B + C_E}(T_P + g_0 T_N)}_{\text{CAR-T killing}} .$$

During the transient phase, the number of effector CAR-T cells is small and, due to the preconditioning therapy, the tumor burden is far below the carrying capacity. Also, $C_E \approx 0$, and the dynamics is approximated by $dT/dt \approx \rho T$, leading to an approximating solution $T = T_0 e^{\rho t}$ with slope $\rho$. During the shrinkage phase, the number of CAR-T cells is very high, leading to $C_E/(B + C_E) \approx 1$ while the majority (if not all) of tumor cells is antigen-positive, $T \approx T_P + g_0 T_N$. Then, $dT/dt \approx \rho T - \gamma T$ and the approximating solution is $T = T_{\text{shr}} e^{(\rho - \gamma)t}$ with slope $\rho - \gamma$. Finally, if the tumor cells were not extinct, the tumor response phase starts during the persistence phase of CAR-T cells with small number of effector CAR-T cells. Therefore, $dT/dt \approx \rho T$ and the approximating solution is $T = T_{\min} e^{\rho t}$ with slope $\rho$.

**Model setup**  Here we describe details of model setup, initial conditions, and unit conversion. In instances where the initial tumor burden was either unspecified (NA) or had 0% blasts, we established a baseline of $T_P(0) = 10^7$ cells. Data points where %blasts were approximately 0 after therapy were not included in calibration due to the high uncertainty associated with digitization. For patients in the studies by Ma et al. [32] and Li et al. [55], where tumor burdens were expressed as percentages, we made an approximation based on the assumption that this percentages are related to white blood cells (WBC) and an adult has 5L of blood. Using an WBC range of $(2.0 - 6.5) \times 10^{10}$ provided in [68], we have an average of $4.25 \times 10^{10}$ of WBC cells. The absolute number of tumor cells was then estimated using the formula: tumor cells = WBC cells × % blasts.

Despite an extensive literature search for datasets with patient time courses sufficient for CAR-T multiphase and IL-6 kinetics, there was insufficient data to describe the phenotypic

characterization of the CAR-T product upon infusion. Consequently, we assumed the administered dose of CAR-T cells as the initial condition of the injected CAR-T cell population ($C_I(0)$). The initial populations of other phenotypes were set to zero ($C_E(0) = C_P(0) = 0$) as these phenotypic changes are driven by antigen-binding and occur throughout the dynamics' evolution. For antigen-negative tumor cells, except for patient G02, the initial tumor burden ($T_N(0)$) was fitted below the detection limit of $2.5 \times 10^5$ cells.

The scaling factor for CAR-T cell count data was estimated using data provided by Lee et al. [69] and Kalos et al. [56], where both counts per microgram and total circulating cells were reported. The conversion factor was determined to be 1 CAR-T cell copy/$\mu$g DNA = $10^4$ CAR-T cells. For conversions between the absolute number of CAR-T cells and cells/$\mu$L, we used the same blood volume of 5L in humans, which gives us 1 CAR-T cell/$\mu$L = $5 \times 10^6$ CAR-T cells. Furthermore, considering that about 1% of cells are present in the peripheral blood (PB) at all times compared to the bone marrow (BM), we have 1 CAR-T cell/$\mu$L in BM = $5 \times 10^8$ CAR-T cells in PB.

To prevent artificial regrowth, we applied zero-limits to CAR-T and tumor cell populations: if any cell population reached less than one cell, proliferation was set to zero.

**Parameter estimation for CAR-T and tumor cells.** A brief evaluation of the multi-layer structure of the model shows that a two-step fitting procedure that considers each layer at a time is equivalent to a one-step approach that fits all model parameters at once. Indeed, since Eqs (1)–(7) for CAR-T and tumor cells are independent of macrophages or IL-6, writing the optimal fit as ($p_1, p_2$), where $p_1$ contains the CAR-T and tumor cell parameters while $p_2$ contains the macrophage and IL-6 parameters, we see that varying $p_2$ does not change the equations (and thus the residuals) of the first layer. Thus, obtaining a global fit ($p_1, p_2$) is equivalent to first optimizing $p_1$ for the CAR-T and tumor cell dynamics (Eqs (1)–(7)) and then optimizing $p_2$ to fit IL6 dynamics (Eqs (8)–(11)).

Therefore, for all 25 patients with CAR-T and tumor cell time courses, we first estimated the 12 parameters for the CAR-T and tumor multiphasic dynamics (Fig 1A, 1B) as follows. First, to reduce the number of free, patient-specific parameters, we analyzed the parameters that least influenced the multiphasic dynamics of CAR-T and tumor cells (S5 Fig), and defined constant, universal values for parameters $K$ (carrying capacity of tumor cells), $\theta$ (recruitment rate of persister CAR-T cells). We then defined reasonable ranges for the other 10 parameters, using literature data when available [4,5,7,70–72] (S2T).

Second, we performed a manual and iterative process to qualitatively fit the remaining 10 patient-specific parameters. This process was based on the analysis that identified a mapping between the 10 shape features and the 10 patient-specific parameters (Figs 4 and S5) and is described as follows. We first analyzed the overall CAR-T cell dynamics on a logarithmic scale, segmenting the data into distinct phases: distribution, expansion, contraction, and persistence. For each phase, we determined an exponential curve, establishing initial estimates and bounds for the dominant parameters $\mu_I, \mu_E, \mu_P$ and $\kappa$ (S11 Fig and S2T). Combining these initial approximations with our knowledge of the influence of the other parameters ($\eta, \epsilon, A, B, \gamma, \rho$) in each shape feature of the model solutions (Fig 4), we manually obtained the initial estimates for the 10 patient-specific parameter values.

Finally, these initial estimates were further improved by an appropriate optimization algorithm that guarantees local minima. To perform this step, we encoded the model structure in QSP Designer [73] and employed the Nelder-Mead method for parameter estimation, minimizing the weighted least squared error (WLS) between model simulations and data. Weights were defined as the inverse of the square of the maximum observed values. After initial runs,

we refined the parameter ranges, and this iterative process continued until an adequate fit to the overall CAR-T cell dynamics was achieved.

The same approach was applied to the extended model for patients with antigen-negative tumor relapse, Eqs (6)–(7), with an extra free parameter $g_0$ (fraction of citotoxicity reduction for antigen-negative tumor cells), which we estimated to be a small fraction [7].

With this process, we obtained the fits for CAR-T and tumor cells shown in Figs 2 and S1–S3. These values are provided in S3 Table.

**Parameter estimation for IL-6 dynamics.** After fitting the model parameters for the first layer encompassing CAR-T and tumor cells, we fitted the macrophage and IL-6 parameters using IL-6 time-courses of 15 patients.

Eqs (8)–(11) have 12 parameters, namely $\sigma_M, \delta_M, \sigma_I, \delta_I, \alpha, \beta_B, \beta_K, \beta_C, C, M_i(0), M_a(0)$ and $IL_6(0)$. After initial analysis of the model dynamics using parameter values in reasonable ranges, we fixed the following 4 parameters as follows. The initial number of activated macrophages was set to zero, $M_a(0) = 0$. The initial condition for IL6 was obtained from the first time point for each patient, $IL_6(0) = IL_{6_1}$. In absence of IL-6 release by activated macrophages, IL-6 levels reach the steady state $\sigma_I/\delta_I$, which is therefore a lower bound for the minimum IL-6 level during the response phase, thus we set $\sigma_I = \delta_I \min_i IL_{6_i}$ for each patient. The model dynamics did not present high sensitivity to saturation parameter $C$, so it was also fixed; since the range for the initial number of inactive macrophages $M_i(0)$ was estimated between $[10^9, 10^{11}]$ cells [74–76], we set the value $C = 10^{10}$ to describe a saturation within the range of inactive macrophages. For the 8 remaining parameters we defined biologically reasonable ranges based either on literature estimates or steady state conditions [22,77–80], with values provided in S2 Table.

Using the defined parameter ranges, we applied an automated routine coded in *Mathematica* and based on the strategy used in [81], combining global and local optimization methods to estimate patient-specific parameter values. First, for each patient, it performs a global search using a Monte Carlo approach, consisting of simulating the model for $10^4$ randomly chosen parameter tuples within these ranges. For each simulation, the logarithm of the weighted sum of squares is computed as

$$LR = \log\left(\sum_{i=1}^{n_d} w_i \left(IL_6(t_i) - IL_{6_i}\right)^2\right), \tag{12}$$

where $(t_i, IL_{6_i})$ are the $n_d$ individual time points. To capture the IL-6 peaks, we used weights given by $w_i = w_0 + (1 - w_0) IL_{6_i} / \max_j(IL_{6_j})$. A value $w_0 = 0.15$ was chosen after initial tests, giving a weight approximately seven times higher to the IL-6 peak in comparison with the minimum value of $IL_{6_i}$.

The routine then selects the 100 parameter tuples that give the minimum $LR$ and refines these estimates by applying to each one a local minimization approach. The best fit is then selected as the one that gave the minimum $LR$. Using this approach, we obtained the fits shown in Figs 2 and S4, which were used in the Results Section. Pairwise correlations between the 100 refined best fits are shown in S12 Fig.

**Assessing different sources of macrophage activation.** To asses the different sources of macrophage activation, we split the macrophage population in three sub-populations according to the activation mechanism, $M_a = M_{a,1} + M_{a,2} + M_{a,3}$, where $M_{a,1}$ are the DAMP-activated

macrophages, $M_{a,2}$ are the antigen-binding-activated macrophages and $M_{a,3}$ are the CD40-activated macrophages. The differential equations for each population are obtained by splitting the activation rate (11), leading to

$$\frac{dM_{a,j}}{dt} = h_j M_i - \delta_M M_{a,j}, \quad j = 1, 2, 3,$$

(13)

where the activation of each sub-population is given by

$$h_1 = \beta_K \frac{C_E}{B + C_E}(T_P + g_0 T_N), \quad h_2 = \beta_B \frac{T_P}{A + T_P} C_E, \quad h_3 = \beta_C \frac{M_a}{C + M_a} C_E.$$

(14)

With this, we calculated for the simulated solutions the peak time for each macrophage population. Further, the total number of macrophages activated by each mechanism is obtained by evaluating the integrals $\int_0^{t_F} h_j M_i dt$, $j = 1, 2, 3$ until a final time $t_F$.

**Model comparison.** To test the robustness of our conclusions on the timeline of macrophage activation, and to determine the relative importance of each mechanism in driving IL-6 dynamics, we compared the full model with alternative, reduced model structures. Denoting the model mechanisms as D (DAMPs release), A (antigen-binding) and C (CD40 contact), we compared the full model (denoted DAC model) with the reduced models DC, AC, DA, D and A, each corresponding to the removal of one or two mechanisms at a time and corresponding to setting $\beta_K$, $\beta_B$ or $\beta_C$ to zero. The model with CD40 contact only (model C) was not tested as it does not result in macrophage activation at all since the CD40 axis requires the presence of previously activated macrophages (see Eq (9) with $\beta_K = \beta_B = 0$). For each alternative model, we repeated the same automated parameter estimation procedure described above and determined the best fit for each patient. For each model and each patient $j$, we calculated the logarithm of residual sum of squares $LR_j$ (Eq (12)) and the AIC criteria, given by

$$AIC_j = 2k + n_{d,j}(1 + LR_j - \ln n_{d,j} + \ln 2\pi),$$

(15)

where $k \in \{6, 7, 8\}$ is the number of free parameters and $n_{d,j}$ is the number of data points of patient $j$. The full DAC model had the minimum sum of $LR_j$ and $AIC_j$ and was the best model for 11 out of 15 patients, followed by models DC in second place and AC in third, both with $LR_j$ and $AIC_j$ in the same range (S8 Fig). Interestingly, both AC and DC models assume CD40 contact as an activation mechanism, differing only on the triggering mechanisms (antigen-binding in the AC model and DAMPs release in the DC model). The analysis performed for the DAC model in Fig 7 was repeated for the reduced AC and DC models and resulted in the same timeline for macrophage activation as well as an equal or higher percentage of CD40 activated macrophages (S9 Fig). Simulating a 50% reduction in each activation mechanism also led to similar reductions in IL-6 peak (compare Figs 8A and S9). These results show that at least two mechanisms are required to accurately describe the dynamics of IL-6, with CD40 contact implied to be the main driver of CRS.

## Supporting information

**S1 Fig. Model simulations for responders.** Model fits for selected patients that showed either complete response (CR) or partial response (PR). CR is achieved when the tumor is clinically undetectable, while PR is defined by a final tumor burden at least less than 50% of the initial

burden. The CAR-T cell detection threshold of $2.5 \times 10^5$ cells is represented by the gray shaded area.
(TIF)

**S2 Fig. Model simulations for patients with antigen-positive relapse.** Model fits for selected patients that showed relapse of antigen-positive tumor cells, excluding data points where %blasts approached zero. Tumor cell error bars represent the range of WBCs used in scaling (see Methods). The CAR-T cell detection threshold of $2.5 \times 10^5$ cells is represented by the gray shaded area.
(TIF)

**S3 Fig. Model simulations for patients with antigen-negative relapse.** Model fits for selected patients that showed relapse of antigen-negative tumor cells. Except for patient G02, the initial tumor burden ($T_N(0)$) was fitted below the detection limit of $2.5 \times 10^5$ cells (gray shaded area).
(TIF)

**S4 Fig. Model simulations of IL-6 kinetics for selected patients**. Experimental data points (orange dots) from [1,32,55,57] are compared with model predictions (orange line). When data was presented data as serum fold change, we establish a direct relationship by considering either a baseline value of 1 ng/L [1] or the specific baseline values provided for each patient [56,57]. The mean values within the reported range for dataset [32] are visually represented by bars in the figure.
(TIF)

**S5 Fig. Sensitivity analysis**. A systematic sensitivity analysis identifies the effect of each of the ten basic mechanistic parameters on the dynamics of CAR-T and tumor cells. The reference simulation is shown in black and alternative scenarios are shown in blue and yellow, where one parameter is changed at a time.
(TIF)

**S6 Fig. Sensitivity analysis**. The relative change in **A** number of CAR-T cells at peak ($C_{max}$) (all patients) and **B** minimum tumor load (non-responders) achieved during the shrinkage phase ($T_{min}$) were calculated as each parameter varied $\pm 10\%$ at a time. Yellow and blue bars indicate the maximum and minimum percentage changes, respectively. Upward triangles indicate the % change for a 10% increase, while downward triangles indicate the % change for a 10% decrease in each parameter value.
(TIF)

**S7 Fig. Effect of different dosing protocols on CAR-T cell dynamics and tumor response.** Comparing the standard scenario (Fit) with simulations starting with either a different CAR-T dose or initial tumor burden (10x and 100x smaller and higher). Assessed outcomes: **A** number of tumor cells at day 90, **B** number of persister CAR-T cells at day 90.
(TIF)

**S8 Fig. Model comparison.** The automated approach used to estimate macrophage and IL-6 parameters in the model considering three activation mechanisms (denoted DAC - Damps, Antigen, CD40) was applied to the reduced models considering 2 or 1 activation mechanisms (DA, DC, AC, D, A), obtained by setting some $\beta_i = 0$. Model C was not considered because CD40 activation depends on the presence of previously activated macrophages. Panels **A** and

**B** show for each model the sum of LR$_j$ (Eq (14)) and AIC$_j$ (Eq (15)) over all patients. Panel **c** shows the model ranking among patients.
(TIF)

**S9 Fig. Simulation results for reduced models AC and DC. A,C** Timelines of macrophage-activation, IL-6 and CAR-T cell peaks for each patient. The percentage of CD40-activated macrophages when all patients are combined was 55% and 75% in models AC and DC, respectively. **B,D** Simulation results of interventions reducing mechanisms of macrophage activation one at a time, showing the the overall reduction in IL-6 peak. See Figs 7 and 8 for further details.
(TIF)

**S10 Fig. Sensitivity analysis.** A local sensitivity analysis was performed to assess the effect of varying parameters on the % reduction in IL6 peaks. For each patient, a constant 50% reduction in each activation mechanism was simulated and the mean reduction in IL6 peak was calculated (reference scenario, black dots, same values as shown in Fig 8A fourth panel); this analysis was then repeated by increasing (+) and decreasing (−) each parameter in 50% increments.
(TIF)

**S11 Fig. Segmentation of CAR-T cell phases.** Segmentation of CAR-T multiphasic dynamics was performed by defining an exponential curve for each phase. For the distribution phase, we consider data points ranging from the dose until the minimum level of CAR-T cells observed, until 5 days. For the expansion and contraction phases, we consider all data points within the endpoints, which mark the distribution and persistence phases. If the persistence phase is not well marked, we choose the slope that best describes the interior points. Finally, the distribution phase is defined until the last observed data point.
(TIF)

**S12 Fig. Pairwise correlations for patient-specific IL-6 best fits.** Each plot shows the pairwise correlations between all parameter pairs, considering the 100 best fits obtained after parameter fitting for the IL-6 model, see Methods for details. The median of all pairwise correlations is also shown.
(TIF)

**S1 Table Overview of patient data**. Individual CAR-T cell dose, the end of analysis, disease, outcome, initial tumor burden and baseline IL-6 concentration. In the split-dose regimen, the CAR-T cell dose was given in 3 fractions with 10% administered on day 0, 30% on day 1, and the remaining 60% on day 3.
(TIF)

**S2 Table Parameter estimation**. Model parameters, their biological meanings, bounds, and references used for parameter estimation.
(TIF)

**S3 Table Parameter values**. Patient-specific fitted parameters. Parameters whose values were the same for all patients are: $C_T(0) = C_P(0) = M_a(0) = 0, K = 4.25 \times 10^{12}$ cells and $\theta = 10^{-8}$ day$^{-1}$.
(TIF)

## Author contributions

**Conceptualization:** Daniela S. Santurio, Luciana R. C. Barros, Ingmar Glauche, Artur C. Fassoni.

**Data curation:** Daniela S. Santurio.

**Formal analysis:** Daniela S. Santurio, Artur C. Fassoni.

**Investigation:** Daniela S. Santurio, Artur C. Fassoni.

**Methodology:** Daniela S. Santurio, Artur C. Fassoni.

**Software:** Daniela S. Santurio, Artur C. Fassoni.

**Supervision:** Artur C. Fassoni.

**Validation:** Daniela S. Santurio, Artur C. Fassoni.

**Visualization:** Daniela S. Santurio, Artur C. Fassoni.

**Writing – original draft:** Daniela S. Santurio, Luciana R. C. Barros, Ingmar Glauche, Artur C. Fassoni.

**Writing – review & editing:** Daniela S. Santurio, Luciana R. C. Barros, Ingmar Glauche, Artur C. Fassoni.

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
