## [Decision Letter · Decision Letter 0]

5 Dec 2024

PCOMPBIOL-D-24-01476

Mathematical modeling unveils the timeline of CAR-T cell therapy and macrophage-mediated cytokine release syndrome

PLOS Computational Biology

Dear Dr. Fassoni,

Thank you for submitting your manuscript to PLOS Computational Biology. After careful consideration, we feel that it has merit but does not fully meet PLOS Computational Biology's publication criteria as it currently stands. Therefore, we invite you to submit a revised version of the manuscript that addresses the points raised during the review process.

Please submit your revised manuscript within 60 days Feb 04 2025 11:59PM. If you will need more time than this to complete your revisions, please reply to this message or contact the journal office at ploscompbiol@plos.org. Please include the following items when submitting your revised manuscript:

We look forward to receiving your revised manuscript.

Kind regards,

David Basanta Gutierrez

Academic Editor

PLOS Computational Biology

Pedro Mendes

Section Editor

PLOS Computational Biology

Feilim Mac Gabhann

Editor-in-Chief

PLOS Computational Biology

Jason Papin

Editor-in-Chief

PLOS Computational Biology

**Journal Requirements:**

1) We ask that a manuscript source file is provided at Revision. Please upload your manuscript file as a .doc, .docx, .rtf or .tex. If you are providing a .tex file, please upload it under the item type LaTeX Source File and leave your .pdf version as the item type Manuscript.

3) We have noticed that you have uploaded Supporting Information files, but you have not included a complete list of legends. Please add a full list of legends for your Supporting Information files after the references list.

Potential Copyright Issues:

i) Figure 1. Please confirm whether you drew the images / clip-art within the figure panels by hand. If you did not draw the images, please provide (a) a link to the source of the images or icons and their license / terms of use; or (b) written permission from the copyright holder to publish the images or icons under our CC BY 4.0 license. Alternatively, you may replace the images with open source alternatives. See these open source resources you may use to replace images / clip-art:

5) We noticed that the Data Availability statement mentioned in the manuscript is different from that provided in the submission form. The submission form states the following "All relevant data are within the manuscript and its Supporting Information files." The manuscript statement; however, states "Upon acceptance, these will be made publicly available in an appropriate repository." Please provide a complete Data Availability Statement in the submission form, ensuring you include all necessary access information or a reason for why you are unable to make your data freely accessible.

Please note that, though access restrictions are acceptable now, your entire data will need to be made freely accessible if your manuscript is accepted for publication. This policy applies to all data except where public deposition would breach compliance with the protocol approved by your research ethics board. If you are unable to adhere to our open data policy, please kindly revise your statement to explain your reasoning and we will seek the editor's input on an exemption. Please be assured that, once you have provided your new statement, the assessment of your exemption will not hold up the peer review process.

7) Please ensure that the funders and grant numbers match between the Financial Disclosure field and the Funding Information tab in your submission form. Note that the funders must be provided in the same order in both places as well. Currently, the grant recipients do not match in both places.

Please indicate by return email the full and correct funding information for your study and confirm the order in which funding contributions should appear. Please be sure to indicate whether the funders played any role in the study design, data collection and analysis, decision to publish, or preparation of the manuscript.

**Reviewers' comments:**

Reviewer's Responses to Questions

**Comments to the Authors:**

**Please note that the review is uploaded as an attachment.**

Reviewer #1: In this manuscript, the authors propose a new mathematical model of

CAR-T cell cancer therapy that includes CAR-T cell dynamics of expansion

and memory formation, antigen-negative resistance, and

macrophage-associated cytokine release. Cytokine release syndrome (CRS)

can be a severe side-effect of CAR-T cell therapy. The model was

calibrated on a dataset of 25 patients collected from published

literature.

The model was built on three different layers: CAR-T cell dynamics,

tumor cell dynamics, and macrophage dynamics. The model consists of

eight ordinary differential equations describing CAR-T cell subsets

(three equations), tumor subpopulations (two equations), macrophage

subsets (two equations) and IL-6 release (one equation). Model inference

was made possible by fitting parameter layer-wise, first CAR-T cell and

tumor dynamics, then macrophage activation and IL-6 release.

The manuscript focuses on the mechanisms leading to macrophage

activation and the ensuing IL-6 release following CAR-T injection. Three

macrophage activation mechanisms were investigated: DAMPs release

(molecules released by dying tumor cells), CAR-T cell antigen

recognition, and cell-cell contact though CD40-CD40L interaction. Using

IL-6 release as a proxy for CRS, the contribution of each mechanism to

CRS was quantified. CD40/CD40L interaction was identified as the most

important factor. Mitigation strategies were searched by sensitivity

analysis. CAR-T cell doses, or preconditioning treatments that reduce

initial tumor burden could decrease the risk of severe CRS.

Pharmaceutical inhibition of macrophage activation soon after CAR-T

infusion could also prevent cytokine release.

The manuscript presents several interesting results. First, the

collected dataset is by itself of interest, and the authors show that

primary clinical parameters are not necessarily good predictors of IL-6

release. Second, by decomposing the contribution of each macrophage

activation mechanism, the authors show that the mechanisms are

temporally distinct. This in turn is helpful for setting up strategies

for limiting cytokine release syndromes. Overall, this study shows how

combining data analysis and mathematical modeling can be used to propose

better treatment combinations/protocols for mitigating severe side

effects of CAR-T therapies.

Comments

1 - The weakest link of this study is the assumption that IL-6 release

is a strong indicator of severe CRS. The authors fail to directly

address this assumption, except in passing (lines 105-107, 111-114).

Yet, one of the main conclusion is that ... our modular model structure accurately describes ... the occurrence of macrophage-mediated CRS.

(lines 125-128). The link between IL-6 and the risks of severe CRS

should be discussed and backed-up more clearly.

2 - The sensitivity analysis is local only (one parameter at a time).

Did the authors consider performing a global sensitivity analysis (Sobol

sensitivity analysis, Fourier amplitude sensitivity test…) to get a

broader quantification of sensitivity outside the best-fit parameter

values.

Lines 173, 181: parameters should be described when first introduced in

the text.

Line 177: ‘such as’ should be removed, the description lists all four

parameters.

Line 182: Does the sensitivity analysis concern all parameters or only a

subset? Please clarify.

Lines 187-190: As they are reported, it looks like all measured

responses depend on all parameters. Perhaps put the results into

perspective by indicating also which response/parameter pairs are

independent?

Section starting on line 216: As far as I can see, this section is

devoted to data analysis only, and is independent from the mathematical

model. It is unclear why modeling is referred to (on lines 222-223).

In the same section, correlation results are not supported by

statistical analysis. Basic regression analysis should be provided,

along with relevant statistics (n, p-values, R^2, F values).

Moreover, on lines 231-234, conclusions are drawn on CRS, no data on CRS

is provided.

Line 235: ‘unveiled’. Is this analysis original? If so, it should be

emphasized, or put into perspective with related clinical observations.

Line 245. ‘Neglecting the patients who achieved sustained remission…’

This clearly poses statistical difficulties. Some of the strong

responders have low CAR-T fold changes, so it is not clear why these

patients can be excluded.

Line 271-273: ‘This suggests that smaller CAR-T cell doses…’ It is not

immediately clear if the modeling results are being interpreted in real

clinical setting, i.e. that ‘smaller CAR-T cell doses extend the

expansion phase…’ is a model prediction, or if the extension of the

expansion phase is the proposed interpretation of the observation that

‘small doses resulted in decreased IL-6 fold change and delayed CAR-T

cell and IL-6 peaks’.

Line 287: ‘It rather appears that a saturation arises…’ Please clarify

what is meant by ‘saturation’; I don’t see it from Fig 6d alone.

Lines 293-295: The sentence seems redundant here, as the results are

reported two paragraphs above.

Line 301-302 ‘CRS timeline’ could be replaced by ‘IL-6 release/fold

change timeline’, unless compelling evidence that CRS = IL-6 fold

change.

Line 312: ‘beta_B, beta_C = 0’ -> ‘beta_B = 0, beta_C = 0’. Similar

comments for lines 313, 315 and 318.

Line 373: Wang et al (2022) https://doi.org/10.3389/fimmu.2022.814548

state that severe CRS appear in average two days post-CAR-T cell

infusion (earlier that mild CRS). How does the proposal that blocking

macrophage activation 1-3 days post-infusion compare with these

observations? More generally, if CRS appear soon after infusion, then

CD40-mediated macrophage activation, which occurs late, would not be the

main driver for CRS. This should be discussed.

Lines 454-479 discusses the framework used for CAR-T cell activation

paths. While this discussion is an interesting topic, it is not central

to this manuscript. Unless the modeling choices for CAR-T cell fate

decision are important for the results (I guess the author assume that

their results are relatively robust), this section could be shortened.

Equations

I have not checked the equations thouroughly, but the simulation results

seem in agreement with the model description.

Reviewer #2: The review is uploaded as an attachment.

**Have the authors made all data and (if applicable) computational code underlying the findings in their manuscript fully available?**

Reviewer #1: Yes

Reviewer #2: Yes

PLOS authors have the option to publish the peer review history of their article (what does this mean?). If published, this will include your full peer review and any attached files.

Reviewer #1: **Yes: **Samuel Bernard

Reviewer #2: No

**Figure resubmission:**
---

## [Decision Letter · Decision Letter 1]

24 Feb 2025

Dear Dr. Fassoni,

We are pleased to inform you that your manuscript 'Mathematical modeling unveils the timeline of CAR-T cell therapy and macrophage-mediated cytokine release syndrome' has been provisionally accepted for publication in PLOS Computational Biology.

There are some comments from one of the reviewers that might improve the manuscript and I encourage you to take them into account.

Best regards,

David Basanta Gutierrez

Academic Editor

PLOS Computational Biology

Pedro Mendes

Section Editor

PLOS Computational Biology

Reviewer's Responses to Questions

**Comments to the Authors:**

Reviewer #1: The authors have significantly improved the manuscript, and have addressed my main concerns. Although there still could be improvements to the manuscript, I do not see any issues that would preclude publication.

Reviewer #2: see attachment.

**Have the authors made all data and (if applicable) computational code underlying the findings in their manuscript fully available?**

Reviewer #1: Yes

Reviewer #2: Yes

PLOS authors have the option to publish the peer review history of their article (what does this mean?). If published, this will include your full peer review and any attached files.

Reviewer #1: **Yes: **Samuel Bernard

Reviewer #2: No

---

## [Editor Report · Acceptance letter]

PCOMPBIOL-D-24-01476R1

Mathematical modeling unveils the timeline of CAR-T cell therapy and macrophage-mediated cytokine release syndrome

Dear Dr Fassoni,

I am pleased to inform you that your manuscript has been formally accepted for publication in PLOS Computational Biology. Your manuscript is now with our production department and you will be notified of the publication date in due course.

With kind regards,

Zsofia Freund
